# Dissecting the contributions to non-photochemical quenching in a land plant under fluctuating light

Lam Lam[1,2,10], Rebecca Lee [3,10], Dhruv Patel-Tupper [2,4], Henry E. Lam [5], Tsung-Yen Lee[2,5], Alexa Ma[5], Sophia A. Ma[4], Hetty He[6], Krishna K. Niyogi [2,4,7,8] & Graham R. Fleming [1,2,5,9] ✉

Photosynthetic organisms have evolved multiple non-photochemical quenching (NPQ) processes, providing photoprotection by safely dissipating excess excitation energy. These processes involve various molecular players functioning on overlapping timescales from seconds to days, making it challenging to isolate and quantify their individual kinetics. In this study, we perform whole-leaf chlorophyll fluorescence lifetime and xanthophyll concentration measurements on wild-type and various newly characterized NPQ mutants of *Nicotiana benthamiana*, a vascular land plant. Based on these measurements, we construct a fluorescence lifetime-based quantitative kinetic model that disentangles individual photoprotection components and, when integrated additively, accurately predicts wild-type and mutant NPQ behaviors under various light-dark regimes. Additionally, the model quantifies the per-molecule quenching effectiveness of various xanthophylls and the contributions of six quenching components (qE$_V$, qE$_A$, qE$_Z$, qE$_L$, qZ, and qI) across different genotypes. It also suggests improved overall quenching efficiency at specific VDE:ZEP:PsbS overexpression stoichiometries, aligning with previous studies and supporting translational efforts to optimize photoprotection and enhance crop yields under dynamic light environments.

The initial step in photosynthesis involves the absorption of a photon and excitation of chlorophyll (Chl) to its singlet excited state ($^1$Chl*) within light-harvesting complexes (LHCs), facilitating electronic energy transfer to the reaction center (RC) to drive charge separation[1]. However, under high light (HL) conditions, the photosynthetic electron transport chain can become saturated. When $^1$Chl* lacks an adequate electron acceptor, formation of the longer-lived triplet state of chlorophyll ($^3$Chl*) via intersystem crossing can occur[2]. These triplet states, when interacting with molecular oxygen, produce reactive oxygen species (ROS), which pose a threat to thylakoid membrane integrity by damaging essential pigments, proteins, and lipids[3,4]. To prevent or mitigate such damage, oxygenic photosynthetic organisms have evolved multiple sophisticated photoprotection pathways, collectively known as NPQ, that safely dissipate excess excitation energy as heat[5,6].

[1]Graduate Group in Biophysics, University of California, Berkeley, CA, USA. [2]Molecular Biophysics and Integrated Bioimaging Division, Lawrence Berkeley National Laboratory, Berkeley, CA, USA. [3]Department of Chemistry, University of California, Los Angeles, CA, USA. [4]Department of Plant and Microbial Biology, University of California, Berkeley, CA, USA. [5]Department of Chemistry, University of California, Berkeley, CA, USA. [6]Department of Molecular and Cell Biology, University of California, Berkeley, CA, USA. [7]Howard Hughes Medical Institute, University of California, Berkeley, CA, USA. [8]Innovative Genomics Institute, University of California, Berkeley, CA, USA. [9]Kavli Energy Nanoscience Institute, Berkeley, CA, USA. [10]These authors contributed equally: Lam Lam, Rebecca Lee. ✉e-mail: grfleming@lbl.gov

To date, five distinct types of NPQ have been discovered, with various genetic mutants deepening our understanding of land plant NPQ at the molecular level. Among these types of NPQ, energy-dependent quenching (qE) occurs on a seconds-to-minutes timescale and accounts for the majority of NPQ induction and relaxation in response to dynamic light intensity changes[7]. qE involves the coordinated action of a pH-sensing protein, photosystem II subunit S (PsbS/NPQ4)[8,9], and xanthophyll carotenoids (Cars), primarily zeaxanthin (Zea, Z) and lutein (Lut, L). Under HL conditions, violaxanthin de-epoxidase (VDE/NPQ1) converts violaxanthin (Vio, V) to antheraxanthin (Ant, A) and subsequently to Zea, a process reversed by zeaxanthin epoxidase (ZEP/NPQ2) under low light/dark conditions[10,11]. Lut, produced via lycopene epsilon cyclase (LUT2), independently and additively contributes to NPQ along with Zea[12,13], the relative contributions of each Car to qE are still extensively debated[11,12,14,15].

The longer-lasting Zea-dependent quenching (qZ) functions independently of PsbS, which is required for qE[16]. Unlike qE and qZ, which involve specific Cars, state transitions in light-harvesting complex II (qT) regulate light harvesting by redistributing excitation energy between photosystem I (PSI) and photosystem II (PSII)[17]. Prior tests have shown negligible qT-associated changes in ¹Chl* fluorescence lifetimes under short-term, fluctuating light conditions[18], likely because qT does not directly quench ¹Chl* and thus contributes minimally to the PSII Chl $a$ fluorescence quenching in the organism studied here, *Nicotiana benthamiana*, as supported by prior tests showing negligible qT-associated changes in ¹Chl* fluorescence lifetimes under our experimental conditions[18]. The most recently discovered chloroplast lipocalin-dependent sustained antenna quenching (qH) has, to date, only been detected under long-term cold and HL conditions[19–21], and is therefore also unlikely to be relevant under rapid, fluctuating light conditions. The most slowly relaxing NPQ component, photoinhibition (qI), occurs when photoprotection is insufficient, resulting in the hours to days long photoinactivation of PSII[22]. Collectively, these NPQ processes involve diverse molecular players that function either additively or independently across overlapping timescales to acclimate to varying illumination conditions in nature. This makes it challenging to disentangle and study their individual kinetics and contributions in vivo[23].

The specific molecular interactions and mechanisms involved in these NPQ processes are still extensively debated. In a recent review, Bassi and Dall'Osto[6] describe four distinct mechanisms that involve either Chl–Chl or Chl–Car interactions, where the role of the Cars may be direct[24–26] or as an allosteric effector (e.g., in Chl–Chl interactions)[27–29]. Evidence of the former includes proposed mechanisms suggesting Cars' role as direct quenchers include charge transfer between Chl and Car that quenches Chl $Q_y$[30–33] and excitonic interactions that may produce a low-lying energy level of a Car, which relaxes rapidly to the ground state either directly or via transfer to a Car $S_1$ state[34,35]. Other models propose, however, exclusively support the latter hypothesis where Cars do not function as direct quenchers, but as allosteric effectors that induce conformational changes in LHCs to facilitate energy dissipation[36–38]. Structural and spectroscopic studies support Zea-dependent but structurally mediated pathways contributing to quenching, suggesting that Zea may enable or stabilize specific protein conformations that facilitate NPQ without directly quenching Chl excited states[36,39]. Recent studies have also proposed the formation of PSI-PSII supercomplexes and associated excitation spillover from PSII to PSI as a potential quenching mechanism involving Zea and PsbS[40–42], though its physiological relevance in *N. benthamiana* remains to be fully established.

Given the ongoing debate on the molecular mechanisms (which may all occur with various contributions in parallel), we have constructed a kinetic model for NPQ that is as agnostic as possible to the precise molecular mechanisms while incorporating the known factors of ΔpH activation of VDE and PsbS and the VAZ cycle. The time-dependent contributions of the various actors emerge naturally from the model and may help to advance the discussions of NPQ mechanisms. A complete list of assumptions underlying the construction of this model, including those that were deliberately excluded, is provided in the Supplementary Information (Supplementary Methods).

There have been a number of previous modeling studies of NPQ kinetics, and qE in particular[43–50]. These models have generally been based on the VAZ cycle, with Zea and protonated PsbS acting as the sole quenching-related components, most commonly represented by the 4-state 2-site quenching model proposed by Jahns and Holzwarth[14], which takes into account both open and closed RCs and quenched and unquenched PSII antennas. In the most relevant previous studies[51,52], a xanthophyll cycle-based kinetic model of short-term photoprotection was developed in the alga *Nannochloropsis oceanica*. This model, utilizing Chl fluorescence lifetime and xanthophyll concentration measurements, captured the dynamics of xanthophyll interconversion and quenching activity, and further highlighted Zea as the most effective photoprotective Car, followed by Ant and Vio. The model also predicted the behavior of two mutants that were not included in the parameterization of the model[52]. These findings provided valuable insights into the kinetics of fast-reacting photoprotection and implications for optimizing photosynthetic efficiency.

However, this model had a number of significant limitations: First, the model was based on a definition of NPQ capacity analogous to the generally used definition based on fluorescence intensity ($NPQ = \frac{Fm - Fm'}{Fm'}$ or $NPQ_\tau(t) = \frac{\tau_{dark}(t=0) - \tau(t)}{\tau(t)}$) and, in the latter, calculated using Chl fluorescence lifetimes measured in "snapshots" over time as quenching in excess light or relaxation in darkness occurs. Here $\tau_{dark}(t=0)$ denotes the amplitude-weighted average lifetime of the initial dark snapshot, and $\tau(t)$ denotes the lifetime at the corresponding snapshot sequence time $t$. While convenient for modeling, normalization by the initial dark-acclimated lifetime $\tau_{dark}(t=0)$ forces the assumption of zero initial quenching in the darkness, which may not hold in mutants with impaired NPQ recovery or altered thylakoid protein composition, such as *zep* or *lut2* mutants. This limits the comparability of the $NPQ_\tau$ parameter across different genotypes with differing initial fluorescence lifetimes. Additionally, its dependence on $\tau(t)$ becomes nonlinear at high quenching levels, hence its relationship to true quenching dynamics becomes unclear. In particular, the model assumes that $NPQ_\tau$ is directly proportional to the concentrations of quenching species, which is not true for $NPQ_\tau$ levels > 2. To address this issue, we formulated the model in this study to directly fit the measured Chl fluorescence lifetimes, allowing for a more fundamental and accurate representation of NPQ capacity.

Second, *Nannochloropsis oceanica* lacks specific pigments found in land plants, such as Lut and chlorophyll $b$ (Chl $b$), and it relies on the phylogenetically and functionally distinct algal protein LHCXI, rather than PsbS, to initiate qE[53–55]. Consequently, the model is not directly applicable to land plants. This limitation is particularly significant given the long-standing debate regarding the roles of Zea and Lut as quenchers in qE, as well as their potentially distinct quenching kinetics and contributions[26]. In addition, while the previous model incorporated qZ, it did not include qI—the slowest NPQ component—which is present at substantially higher relative magnitudes in plants but is challenging to disentangle from other NPQ components[23,56]. Therefore, developing a new lifetime-based model tailored to land plants that includes both the Lut-dependent qE quenching pathway and qI is essential to provide deeper insights into these debates and improve our capability to predict and engineer NPQ in plants.

To that end, we selected *N. benthamiana*, an allotetraploid vascular land plant, as our model system due to its compatibility with previous related studies conducted using transient absorption spectroscopic measurements, which facilitate investigation of potential qE quenching mechanisms[26] and assessment of exciton diffusion length in relation to NPQ levels in isolated thylakoids[57]—measurements that are

**Table 1 | Summary of *N. benthamiana* mutants used in this study**

| Genotype | NPQ Phenotype | | References |
|---|---|---|---|
| WT | Wild type | | |
| npq4 | Lacks PsbS | No qE | (Lee et al., 2024)[26] |
| npq1 | Lacks VDE | No Ant or Zea | (Lee et al., 2024)[26] |
| lut2 | Lacks Lut | No qE from Lut | (Lee et al., 2024)[26] |
| zep2 | Reduced ZEP | Accumulates Zea | Generated in this study |
| npq4 npq1 | Lacks VDE and PsbS | No qE or qZ | Generated in this study |
| npq1 lut2 | Lacks VDE and Lut | No qZ, has qE only from Vio | Generated in this study |
| npq4 lut2 | Lacks PsbS and lutein | No qE | Generated in this study |
| npq4 zep2 | Lacks PsbS and reduced ZEP | No qE and accumulates Zea | Generated in this study |
| zep2 lut2 | Reduced ZEP and lacks Lut | No qE from Lut and accumulates Zea | Generated in this study |

challenging to perform in *Arabidopsis thaliana*. This choice could enable both kinetic and mechanistic analysis of NPQ in land plants when combined with complementary techniques across a series of studies. In the present study, however, to preserve physiological function as much as possible, we utilized fresh, intact leaves of *N. benthamiana* to systematically investigate the NPQ response using a comprehensive set of single and double NPQ mutants[26] (Table 1) across three regular/irregular actinic light sequences. By combining five critical mutants, each lacking specific components of the NPQ pathways, we construct a lifetime-based quantitative kinetic model of NPQ focused on the largest components in land plants: qE, qZ, and qI. Model parameters are directly fitted to the experimental Chl fluorescence lifetimes, guided by xanthophyll concentrations measured via high-performance liquid chromatography (HPLC), with all measurements performed in intact leaves as described in more detail in the Methods. Parameterized using four single mutants and one double mutant under fluctuating light, this approach enables us to isolate the activation and relaxation kinetics of NPQ as well as the contributions of each NPQ components individually while capturing their interplay—the model accurately predicts NPQ responses in the four single mutants under constant light, as well as the wild type (WT) and four additional double mutants under three different illumination sequences. Furthermore, the model enables comparisons of per-molecule quenching effectiveness of the xanthophylls and further predicts quenching kinetics upon overexpression of VDE, PsbS, and/or ZEP in various stoichiometries. Altogether, these findings highlight potential strategies to modulate key enzyme expression and activity, offering valuable insights for enhancing NPQ efficiency and improving crop productivity[58,59].

## Results

### An NPQ kinetic model based on chlorophyll fluorescence lifetime

We aimed to construct a minimal model of NPQ dynamics in land plants derived from known biochemical processes. Our kinetic model incorporates three key components: a dynamic xanthophyll cycle, Lut activation, and a simplified phenomenological representation of qI. The model incorporates 13 chemical species: free xanthophylls Vio (V)/Ant (A)/Zea (Z), protein-xanthophyll bound complexes PV/PA/PZ, and an additional protein-Lut complex PL, which are activated to be quenching complexes (QX) in light, as well as the key xanthophyll cycle enzymes VDE and ZEP. A phenomenological representation of qI is given by modeling $\alpha_{qI}$, the ratio of damaged RCs to the whole pool. A simplified schematic of this model is shown in Fig. 1A and a detailed description is provided in the Supplementary Methods section.

To address the limitations of our previous *N. oceanica* model described in the Introduction and expand the model to represent land plant NPQ, we present an alternative approach to directly model quenching using the fundamental definition of fluorescence lifetime.

The total fluorescence decay rate, $\tau_F(t)$, is determined by several competing processes, including the non-radiative decay and the intrinsic fluorescence decay of Chl *a*, qE, qZ, and qI. Assuming the quenching rate is linearly dependent on the concentration of quenching species, we define the fluorescence lifetime of Chl *a* via a Stern-Volmer type approach[60] as:

$$\frac{1}{\tau_F} = \kappa_{r,nr} + \kappa_{qE}[QX] + \kappa_{qZ}[Z] + \kappa_{qI}[I] \quad (1)$$

Where $\kappa_{qE}$, $\kappa_{qZ}$, $\kappa_{qI}$ represent the quenching rate constants of qE, qZ, and qI, respectively, and $\kappa_{r,nr}$ accounts for all the other radiative and non-radiative de-excitation processes. While this phenomenological formulation can include multiple alternative quenching mechanisms, it enables effective kinetic fitting of fluorescence lifetimes, and it was chosen for its simplicity, interpretability, and capacity to capture the dynamic contributions of multiple quenching species. Furthermore, since QV, QA, QZ, and QL may exhibit different per-molecule quenching effectiveness, we further distinguish their contributions within qE:

$$k_{qE}[QX] \approx \kappa_{QV}[QV] + \kappa_{QA}[QA] + \kappa_{QZ}[QZ] + \kappa_{QL}[QL] = \sum_X \kappa_{QX}[QX] \quad (2)$$

This yields the final time-dependent expression for the Chl *a* fluorescence lifetime:

$$\tau_F(t) = \frac{1}{k_{r,nr} + \sum_X \kappa_{QX}[QX] + \kappa_{qZ}[Z] + \kappa_{qI}\alpha_{qI}} \quad (3)$$

This formulation allows for direct quantification of quenching rates and per-molecule effectiveness of the individual quenchers while accounting for genotypic variations in $\tau_{dark}(t=0)$. In this model, $\kappa_{qX}$, $\kappa_{qZ}$, and $\kappa_{qI}$ are additional fitting parameters, while $k_{r,nr}$ is calculated directly from $\tau_{dark}(t=0)$ for each mutant. Details of the model implementation and fitting are given in Methods and Supplementary Information (Supplementary Methods).

### Characterization of xanthophyll cycling in novel NPQ mutants of *N. benthamiana*

To systematically disentangle the individual quenching pathways found in *N. benthamiana*, we characterized a series of mutants, each deficient in one or more NPQ components. Three such mutants were characterized previously: the *psbs1psbs2* mutant that lacks qE (hereafter *npq4*), the *lut2-1 lut2-2* mutant that lacks lutein-dependent qE (hereafter *lut2*), and the *vde1 vde2* mutant that lacks Zea-dependent qE and qZ (hereafter *npq1*)[26]. In this study, we generated and characterized single and double CRISPR-generated mutants deficient in one or both ZEP orthologs. ZEP is necessary for the production of Vio, Ant, and neoxanthin, the latter being an important precursor for the

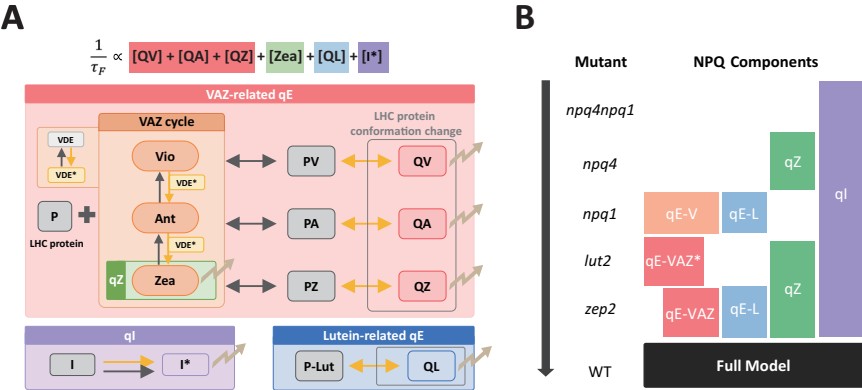

**Fig. 1 | Schematic of the stepwise parameterization of the NPQ model.**
**A** Schematic of the NPQ Model constructed from qE, qZ, and qI components, including the VAZ cycle and lutein activation. The model captures interconversion of Vio, Ant, and Zea and its binding to an LHC protein P (Lut is assumed to be always bound) as indicated by the protein-bound PX state and high light dependent activation to its respective quenching QX state. Here, X is an interchangeable xanthophyll, and the activation of PX to QX is depicted. Arrows represent rate-dependent processes modeled using differential rate law kinetics. Light-independent and light-dependent processes are indicated by black and yellow arrows, respectively. Quenching processes ($\xi$) are modeled using an inverse relation between chlorophyll fluorescence lifetime $\tau_F$ and the amount of each quenching species. **B** Schematic of stepwise training of the NPQ model by decomposition into components using NPQ mutants. The black arrow indicates the sequential fitting of model rate parameters, starting with the simplest mutant (*npq4npq1*) and progressively incorporating additional NPQ components to parameterize the full wild-type (WT) model. Some rate parameters associated with the VAZ-related qE component (in particular, antennae binding) differ in *lut2*, as addressed by *, while other parameters are shared between *lut2* and *zep2*, reflected in the partial overlap in their qE-VAZ components.

stomata-regulating plant hormone abscisic acid[61]. Correspondingly, knockouts of ZEP should constitutively accumulate Zea even in the dark. However, the lack of epoxidized Vio in the *zep1zep2* double knockout in *N. benthamiana* significantly reduced plant viability, with only one of eleven $T_0$ transformants producing viable, stable progeny (Supplementary Table 5). In contrast, both *zep1* and *zep2* mutants were able to grow under ambient greenhouse conditions with more modest differences in growth. While the *zep2* mutant was indistinguishable from WT plants, the *zep1* mutant exhibited stunted growth, frilled leaf edges, and reduced leaf turgor. We found that each ortholog contributes additively to Zea epoxidation, with ZEP1 being the dominant copy. Interestingly, knockout of ZEP2 (hereafter *zep2*) resulted in a moderate knockdown phenotype, with WT-like growth and a near-native Vio and Neo composition and sufficient constitutive Zea to saturate Zea-dependent qE capacity (Supplementary Fig. 1), unlike the non-native composition and slowed, humidity-dependent growth found in the *A. thaliana npq2* point mutant[10]. This *zep2* mutant provided quantitative and temporal variation in Zea-dependent qE and qZ that was necessary and sufficient for phenotypic modeling, without the pleiotropic impacts on xanthophyll composition that are present in existing *A. thaliana* mutants.

The presence or absence of these traits within the qE, qZ, and qI parameters that represent the vast majority of WT land plant NPQ is included in Table 1. These mutants enabled a stepwise model training approach, where parameters were progressively fitted, starting with the simplest NPQ system (*npq4npq1*, which contains only qI) and gradually incorporating additional quenching components to reconstruct the full WT model (Fig. 1B). Whole-leaf pigment profiles were quantified using HPLC under dark-acclimated and HL-acclimated conditions for all genotypes used in this study (Supplementary Fig. 2). WT, *npq4, zep2, lut2,* and *zep2lut2* mutants exhibited robust de-epoxidation of Vio in response to HL, and furthermore, *zep2, npq4zep2* and *zep2lut2* retained more residual accumulation of Ant and Zea in the dark and after HL due to reduced ZEP activity. On the contrary, *npq1* and its double mutants (*npq1npq4, npq1lut2*) displayed no significant changes in pigment profiles before and after HL exposure, indicating a complete loss of VDE function[62–64]. Additionally, *lut2* and its derived double mutants (*npq4lut2, npq1lut2, zep2lut2*) lacked detectable Lut and

exhibited a considerably expanded VAZ pool size, consistent with previous reports[12].

**Training the NPQ model with chlorophyll fluorescence lifetimes**
Whole-leaf Chl fluorescence lifetime measurements were conducted using time-correlated single-photon counting (TCSPC), as described in detail in the Methods section. The genotypes *npq4npq1, npq4, npq1, lut2,* and *zep2* are the minimal set sufficient to fit all parameters, integrating all major photoprotective pathways in order to reconstruct the complete WT model of $\tau_F$ (Fig. 1B). Fluorescence lifetime data was collected using a 5HL-10D-5HL actinic light sequence (Fig. 2A–E). In general, fluorescence lifetimes decreased during the light period, indicating quenching activity, and increased to varying extents in *npq4, npq1, lut2,* and *zep2* during the following dark period, reflecting partial NPQ recovery. Among these, *lut2* and *zep2*, both possessing functional VDE and therefore having Zea as a major contributor to quenching, exhibited shorter lifetimes compared to *npq1*, which lacks Zea. Conversely, *npq4npq1* and *npq4*, both deficient in PsbS, showed less quenching and minimal recovery.

Our model fitting results (Fig. 2A–E) successfully captured the contributions of individual quenching components—*npq4npq1* for qI, *npq4* for qZ, and *npq1* for Vio-dependent qE (qE$_V$) and Lut-dependent qE (qE$_L$) —as well as the combined effects of multiple quenchers (*lut2, zep2*). Except for the initial Vio concentration, which was derived from HPLC results, all fitting parameters remained consistent across mutants. While the qI component was derived from fitting *npq4npq1* lifetime data under the 5HL-10D-5HL sequence (Fig. 2A), an additional control measurement performed using only laser excitation without actinic light on *npq4npq1* (laser baseline) was accurately captured by the qI-only model (Supplementary Fig. 3). This indicates although the laser baseline modestly contributes to detectable quenching, this effect is accounted for in our model within the qI parameters via the positive $\alpha_{qI}$ accumulation rate in darkness in *npq4npq1*, which was modeled phenomenologically due to the unresolved mechanistic basis of qI, yet still offers insight into the temporally resolved contribution of qI.

In *npq4* (Fig. 2B), the relatively slow activation and relaxation dynamics were attributed to the slower qI and qZ kinetics. In *npq1*

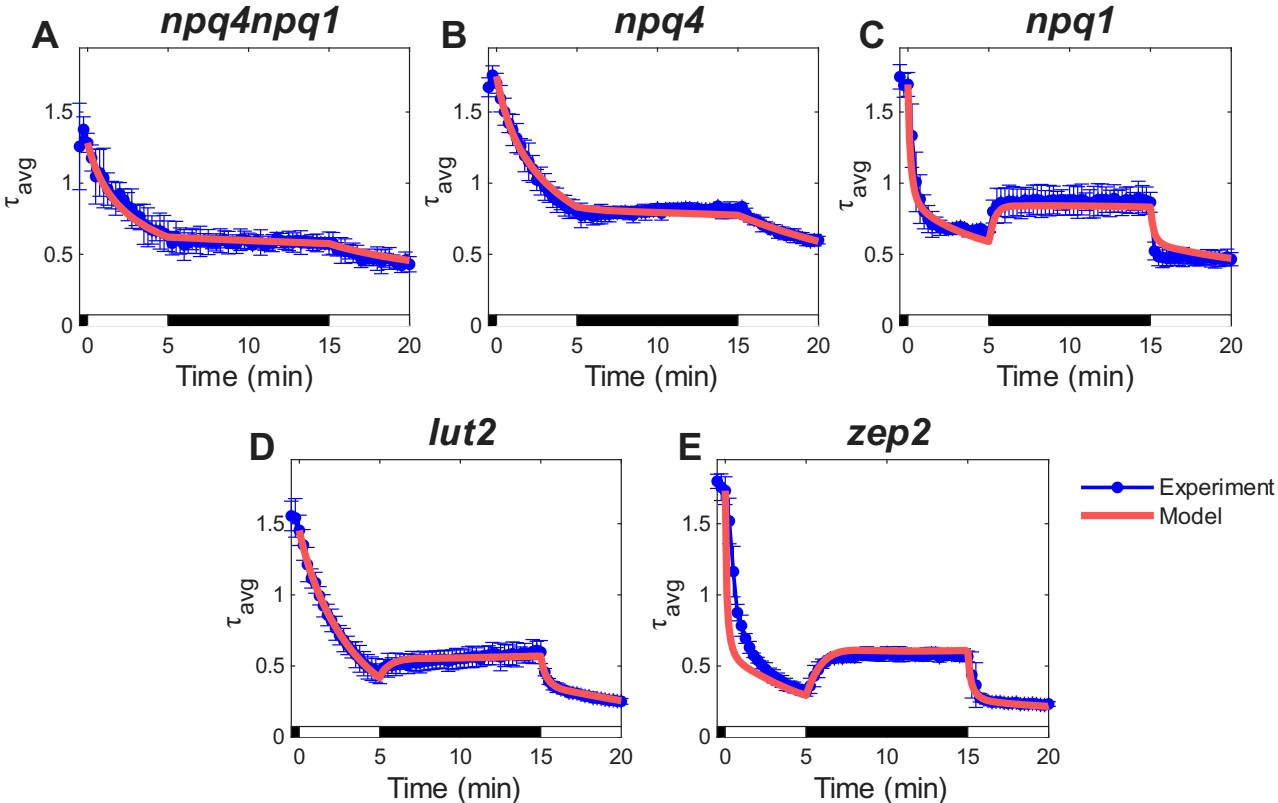

**Fig. 2 | Whole leaf chlorophyll fluorescence lifetime data (blue) and model fitting (red) under the 5HL-10D-5HL sequence for mutants used to train model parameters.** The model was fitted using fluorescence lifetime data of **A** *npq4npq1* (*n* = 3), **B** *npq4* (*n* = 3), **C** *npq1* (*n* = 3), **D** *lut2* (*n* = 3), and **E** *zep2* (*n* = 4) leaves. Model fitting was performed with lifetime data under this sequence together with the 3HL-

1D-1HL-3HL-9D-3HL sequence (Supplementary Fig. 4). RMSD of fittings are (s⁻¹): **A** 0.027, **B** 0.043, **C** 0.125, **D** 0.142, **E** 0.239. White and black bars indicate high light (HL) and dark (D) phases of the actinic light sequence. Error bars represent ± 2 SE from *n* biological replicates.

(Fig. 2C), the NPQ components qI, qE$_V$, and qE$_L$ exhibited rapid activation, contributing to a sharp decrease in lifetime during both the initial and second illumination periods, although the model slightly underestimated Lut activation rates. Partial recovery in the dark period was attributed to the rapid deactivation of qE$_L$. Similarly, *lut2* (Fig. 2D) exhibited substantial contributions to activation from qE$_V$, as well as Ant-dependent qE (qE$_A$), and Zea-dependent qE (qE$_Z$), with lingering qZ causing a prolonged relaxation phase in the dark. In *zep2* (Fig. 2E), the rapid initial NPQ activation was largely driven by pre-existing Zea, leading to a strong qE$_Z$ followed by a slower qZ response. Partial recovery in darkness resulted from qE$_Z$ deactivation, while qZ recovery remained slow. Similar observations and model fitting accuracy were observed under the 3HL-1D-1HL-3D-9HL-3D sequence (Supplementary Fig. 4).

To ensure consistency of rate parameters across genotypes, fluorescence lifetime datasets were fitted in batches. In the absence of Lut, which plays a critical role in maintaining trimeric LHCII stability, *lut2* mutants have altered LHCII pigment compositions (e.g. replacement of Lut by Vio, Ant, and Zea) and xanthophyll binding affinities, which will affect excitation energy quenching[26,64]. As a result, kinetic rate parameters for PX formation specific to *lut2* were used and marked by an asterisk (*). Additionally, to account for genotypic differences in pigment composition, initial pigment concentrations $V_0$ were fitted independently for each genotype.

The kinetic parameters obtained from *npq1*, *npq4*, *lut2*, *zep2*, *npq1npq4*, and WT under HL conditions are summarized in Supplementary Table 1. In *zep2*, the conversion of Zea to Ant and Ant to Vio was reduced because of lower ZEP activity due to knockout of the less dominant of two ZEP paralogs, thus ZEP function was not

entirely eliminated. Large rate constants for the formation of protein-xanthophyll complexes PX, denoted by $X + P \rightleftharpoons PX$, reflect a rapidly equilibrating process, whereas the formation from PX to quenching complexes QX occurred at a much slower rate, influencing NPQ induction dynamics. QL shows a similar per-molecule activation rate to QV but substantially lower than QZ and QA. However, given its high concentration relative to the VAZ pigments, the contribution of QL to overall quenching is nevertheless significant. The rate constants associated with pigment binding and dissociation were found to differ in *lut2*; in particular, the $X \rightleftharpoons PX$ equilibrium shifts to the left for Vio, Ant, and Zea, consistent with reduced LHCII trimer stability, and the ratio of binding affinities between Vio, Ant, and Zea changes from approximately 2:3.5:7 to 1:1:3, consistent with alterations in VAZ binding affinities. The total number of VAZ binding sites, [P]$_{tot}$, is approximately halved in *lut2*, consistent with the importance of Lut as a structural component of the LHCII trimer.

Quenching rate constants for each xanthophyll species are provided in Supplementary Table 2. The quenching rate ($k_X$) for a given species X is defined as $k_X = \kappa_{QX}[X]$, where $\kappa_{QX}$ represents quenching rate constant and [X] represents the concentration of the quenching species (in mmol/mol Chl). These rate constants ($\kappa_{QX}$) reflect intrinsic differences in quenching ability between the pigments once activated. Though $\kappa_{qZ}$ is substantially lower than $\kappa_{QZ}$, [Z] typically exceeded [QZ] by an order of magnitude, so the overall quenching contributions of qZ and qE$_Z$ are similar. The quenching rate constants for Lut and Zea are also similar, with Lut being slightly higher. $\kappa_{ql}$ acts on the fractional quantity $\alpha_{ql}$; thus, the rate of photoinhibition does not exceed 2.82 (mmol/mol Chl⁻¹ s⁻¹) in this model.

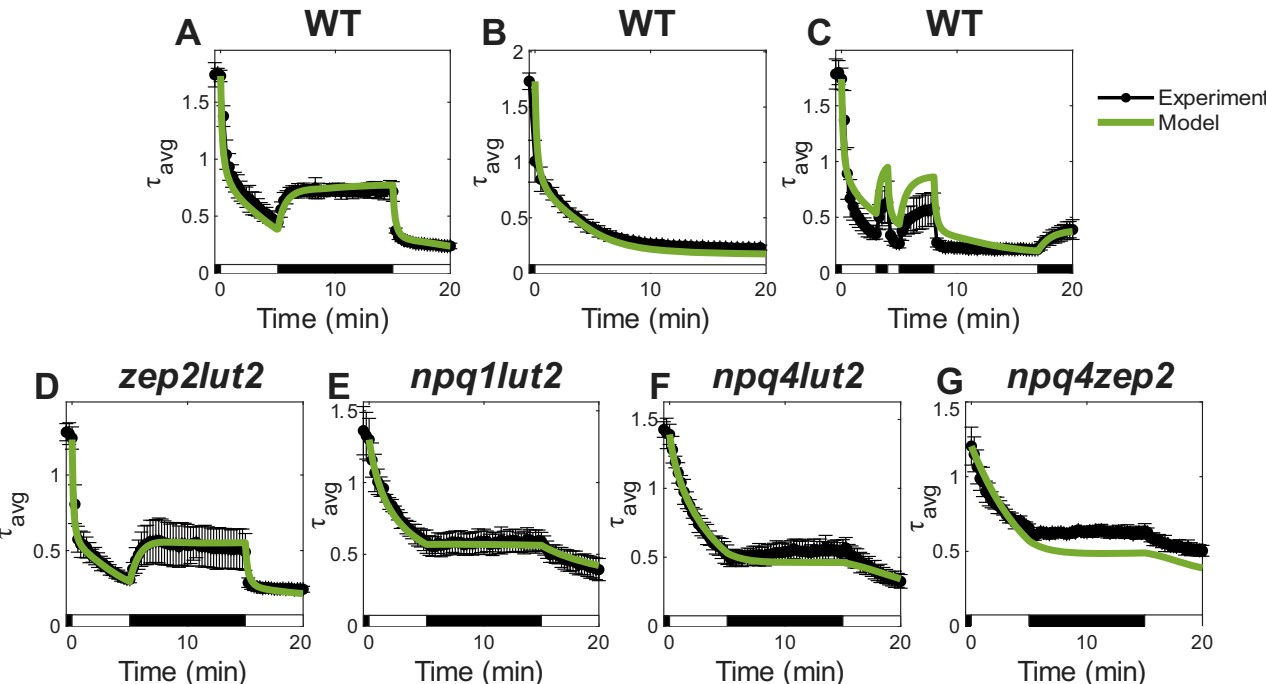

**Fig. 3 | Whole leaf chlorophyll fluorescence lifetime data (black) and model (green) predictions.** Model predicted $\tau_F$ behaviors were generated for WT under three light sequences: **A** 5HL-10D-5HL ($n = 3$), **B** 20HL ($n = 6$), **C** 3HL-1D-1HL-3D-9HL-3D ($n = 3$), and for double mutants under the 5HL-10D-5HL sequence: **D** *zep2lut2* ($n = 5$), **E** *npq1lut2* ($n = 4$), **F** *npq4lut2* ($n = 3$), **G** *npq4zep2* ($n = 5$). Model predictions were calculated from parameters fitted in Fig. 2 and Supplementary Fig. 4, with $[V]_{0,WT}$ fit separately to account for differences in $[V]_0$ observed in HPLC measurements. For double mutants (**D**–**G**), mutant-specific parameters are combined to produce the modeled result. Error bars represent ± 2 SE from $n$ biological replicates. RMSD ($s^{-1}$) = **A** 0.182, **B** 0.935, **C** 0.811, **D** 0.269, **E** 0.023, **F** 0.214, **G** 0.377. White and black bars indicate high light (HL) and dark (D) phases of the actinic light sequence. Error bars represent ± 2 SE from $n$ biological replicates.

## Predictions made by the NPQ model

To assess the predictive capabilities of our kinetic model, we performed fluorescence lifetime measurements on WT and an additional set of *N. benthamiana* mutant leaves under three distinct illumination sequences, as described above. Parameters obtained by fitting the previous five mutants were integrated without modification to construct a complete WT model (only $[V]_{0,WT}$ was fitted separately). Across all illumination sequences, the model successfully captures the overall kinetics of NPQ induction and relaxation in WT leaves (Fig. 3A-C). In particular, the model captures the WT characteristics of slower initial activation of NPQ, partial recovery in dark, and fast activation upon re-illumination. Consistent with our current understanding of *N. benthamiana* photoprotection, the model predicts a strong reliance on qE and qZ under continuous and fluctuating HL conditions, reflecting the dominant role of these components in rapid adjustment to fluctuating light and acclimation to intense light stress.

While the model accurately reproduces NPQ dynamics in the 5HL-10D-5HL and 20HL sequences, it slightly underestimates NPQ activation during the first ~8 min of the irregular 3HL-1D-1HL-3D-9HL-3D sequence. This discrepancy likely arises from biological variability, such as differences in VAZ pool sizes, growth conditions, or leaf-to-leaf variation. Such factors, which may contribute to batch-dependent discrepancies, are further discussed in the Discussion.

Beyond WT, the model effectively captures NPQ dynamics in the four single mutants *npq1*, *npq4*, *lut2*, *zep2* measured under 20-min HL (Supplementary Fig. 5), as well as an additional set of double mutants, including *zep2lut2*, *npq1lut2*, *npq4lut2*, *npq4zep2* in the 5HL-10D-5HL sequence (Fig. 3D–G), with only a slight overestimation of quenching in *npq4zep2* during the later phases of the illumination sequence. This overestimation may result from differences in ZEP levels and total VAZ pool sizes between *zep2* and *npq4zep2* mutants, as indicated by the HPLC pigment profiles discussed above, or it could simply result from

biological variation, as described further in the Discussion. The model also gives reasonable predictions of NPQ behavior in these four double mutants under the 3HL-1D-1HL-3D-9HL-3D sequence (Supplementary Fig. 6). These findings highlight the model's robustness in integrating multiple quenching components.

The agreement between experimental data and model predictions for WT, four single mutants (under 20HL), and four double mutants under different illumination sequences supports the validity of the kinetic parameters obtained from fitting and the model's underlying assumptions. In particular, the model accurately predicts WT induction and relaxation dynamics from a direct sum of the individual quenching pathways associated with $qE_V$, $qE_A$, $qE_Z$, $qE_L$, qZ, and qI, suggesting that the effects of these components can be distinguished from each other and may be considered additive. The generalizability of the kinetic parameters to single mutants (20HL) and double mutants further supports our hypothesis that the contributions to different NPQ components are distinguishable, and that each of the quenching components operates largely independently so that even double knockouts retain the NPQ capabilities from the remaining pathways, although all four qE components ($qE_V$, $qE_A$, $qE_Z$, and $qE_L$) require PsbS. Additionally, the model makes accurate predictions of *lut2* double mutants using the *lut2*-specific PX binding rates, consistent with the effect of Lut on the stability of LHCII trimeric structure[26,64].

## Discussion

NPQ plays a critical role in regulating photosynthetic efficiency and protecting plants from photodamage[7,65]. Here, we present a novel theoretical framework that models NPQ dynamics by directly predicting Chl fluorescence lifetime, with fitting parameters constrained by xanthophyll concentrations. With a series of newly characterized NPQ mutants in *N. benthamiana*, we are able to independently extract Vio/Ant/Zea-dependent qE, Lut-dependent qE, qZ, and qI. Using a

parameterized approach to capture photoprotective dynamics, the model offers predictive insight into both fluorescence lifetime kinetics and xanthophyll concentration in *N. benthamiana*.

In this work, our experimental protocol closes the RCs. We allow each component of the VAZ cycle to act as an inducible contributor to quenching, with differing activation rates and quenching effectiveness. We also include qI and qZ as well as the contributions of Lut, which is an important contributor to the activation and deactivation of qE at short timescales and the major component for the *npq1* mutant also studied by Morales et al.[45]. As Stirbet et al. point out[50], several existing models assume that the VAZ cycle (in particular, Zea) and PsbS protonation are the only components involved in qE quenching (the "4 state 2-site" model), simplifying NPQ dynamics but reducing the ability to capture complex photoprotective systems. These models also commonly make the simplifying assumption that the total pool of [Vio] + [Zea] is constant over short timescales. However, this limits the ability to compare kinetics across genotypes, which can exhibit large variations in the size of the total xanthophyll pool even after dark adaptation (Supplementary Figs. 1, 2). The memory effect explored by Matuszynska et al. is an intrinsic part of our model[43], as it was in our earlier algal model[51,52], and is unraveled in greater mechanistic detail here, demonstrating the importance of the Ant intermediate to NPQ memory during dark periods. Thus, our work here expands upon the existing body of qE modeling studies by providing a detailed molecular picture of photoprotection, connecting lifetime dynamics directly to specific molecular contributors and kinetic rates, and capturing multiple NPQ components (qE, qI, qZ).

The success in reconstructing the WT NPQ response from the individual mutant data implies that the various contributions to overall NPQ are linear and additive. It is also important to note that basing the analysis on fluorescence lifetimes via a Stern-Volmer approach was critical to obtaining transferable parameters. The non-linear relationship of the conventional NPQ definition ($NPQ = \frac{Fm - Fm'}{Fm'}$, or $NPQ_\tau(t) = \frac{\tau_{dark}(t=0) - \tau(t)}{\tau(t)}$) and the variability of the initial dark (RCs closed) fluorescence lifetime would significantly compromise an approach based on conventional NPQ values. Another key assumption of our modeling approach is that the elementary reaction rates underlying each NPQ component remain consistent across different genotypes and that the phenotypic effect of a single mutant remains the same in its corresponding double mutant. While pleiotropic effects of mutations could influence some of these parameters, this assumption is validated by the accurate prediction of most double mutant lifetimes (Fig. 3D−G), and the predictive power of the rates obtained through fitting mutant data suggests a reasonable level of confidence in their values, demonstrating the model's overall robustness.

The kinetic parameters obtained from our model provide a quantitative basis for comparing the relative efficiencies of key NPQ processes across different xanthophylls. Our results indicate that the formation of PX complexes occurs significantly faster than their subsequent activation into quenching states QX, in agreement with previous findings[51,52]. However, comparatively, Zea exhibits the fastest rate of forward PZ formation, as well as the highest equilibrium ratio of bound PZ to free Zea (given by $\frac{k_{pxf}}{k_{pxf} + k_{pxb}}$, where $k_{pxf}/k_{pxb}$ are the forward and backward rates of PX formation, respectively), whereas Vio has the slowest formation and lowest ratio of PV to V (Supplementary Table 1). Although PV/PA/PZ formation is generally slower in *lut2* as discussed previously, the forward PX conversion rates and equilibrium ratios for Vio/Ant/Zea still favor Zea. Similarly, the forward rate of QX activation is highest for Zea. Together, these results suggest that the preference for Zea as an effector of qE quenching originates not only from its inherent molecular quenching ability, but also from its favorability in binding to light-harvesting complexes and forming the active quenching complex, possibly with PsbS. Interestingly, the backward conversion from QX to PX is faster for Zea and Ant compared to Vio,

**Table 2 | Quenching effectiveness of Vio (QV), Ant (QA), and Zea via qE (QZ), Lut (QL), and Zea via qZ (qZ)**

| Species | $Q_{eff}$ |
|---------|-----------|
| QV | 0.0022 |
| QA | 0.0042 |
| QZ | 0.0390 |
| QL | 0.0039 |
| qZ | 0.030 |

Quenching effectiveness was calculated by $Q_{eff} = \frac{k_{pxf}}{k_{pxf} + k_{pxb}} \cdot \frac{k_{qxf}}{k_{qxf} + k_{qxb}} \cdot \kappa_{QX}$, where $k_{pxf}/k_{pxb}$ is the forward and backward rate of PX formation, $k_{qxf}/k_{qxb}$ is the forward and backward rate of QX formation, $\kappa_{QX}$ is the quenching rate of QX, and $\frac{k_{pxf}}{k_{pxf} + k_{pxb}} \cdot \frac{k_{qxf}}{k_{qxf} + k_{qxb}}$ represents the effective maximum fraction of molecules of xanthophyll in a given pool contributing to quenching.

suggesting that QV is both slow to form and slow to deactivate. However, given the extremely low concentrations of QV, the precision of these fitted rates is expected to be lower than that of QA and QZ. In addition, the activation rate of QL is similar to that of QV; however, the considerably higher concentration of lutein allows QL to still form in substantial amounts.

Our model also provides insights into the kinetics of xanthophyll cycle interconversion. Under HL conditions, the estimated rate of Vio-to-Ant conversion is faster than Ant-to-Zea conversion, which appears to contrast with conclusions from the previous *Nannochloropsis* model[52], where A-to-Z conversion was reported as the faster de-epoxidation step. Since no mutant isolates Ant from Zea quenching, fully distinguishing Ant's kinetic contribution from Zea remains challenging. Future studies using engineered enzyme variants could help refine these estimates.

In addition, the model allows for a direct comparison of the per molecule quenching effectiveness for each xanthophyll species to the total amount present, which we define as:

$$Q_{eff} = \frac{k_{pxf}}{k_{pxf} + k_{pxb}} \cdot \frac{k_{qxf}}{k_{qxf} + k_{qxb}} \cdot \kappa_{QX} \qquad (4)$$

where quenching effectiveness is determined by (1) the fraction of pigment X bound to a protein site ($\frac{k_{pxf}}{k_{pxf} + k_{pxb}}$), (2) the maximum fraction of bound protein-xanthophyll complexes PX that activate as quenching complexes QX under light conditions ($\frac{k_{qxf}}{k_{qxf} + k_{qxb}}$, where $k_{qxf}/k_{qxb}$ are the forward and backward rates of QX formation, respectively), and (3) the quenching rate constant of QX ($\kappa_{QX}$). The estimated quenching effectiveness obtained from the WT model are given in Table 2. The model reports Zea in the form of qZ and QZ to be the strongest contributors to quenching and Vio (QV) to be the weakest, while Ant (QA) shows intermediate quenching effectiveness, in line with our current understanding of the relative strength of Vio/Ant/Zea in contributing to NPQ. The significant qZ contributions in our model are consistent with the existence of a light-independent Zea-mediated quenching mechanism[16]. This interpretation is further supported by the observed reduction in the initial $\tau_{dark}$ in *npq4zep2* compared to *npq4*, suggesting a ΔpH-insensitive Zea-dependent quenching component that remains active in the dark-acclimated state in *npq4zep2*. QV, QA, and QZ have an approximately 1:2:18 ratio of quenching effectiveness, a relative increase in the quenching ability of Zea compared to a previous study[52]. This may reflect inherent differences in NPQ mechanistic details between land plants and algae. The per-molecule quenching effectiveness of Lut is similar to Ant. Thus, Zea (both qZ and QZ) generally exhibits a quenching effectiveness 10 times that of Lut, in agreement with previous studies[66]. However, since Lut is generally present in considerably higher concentrations than Zea, the

## NPQ componentwise contributions

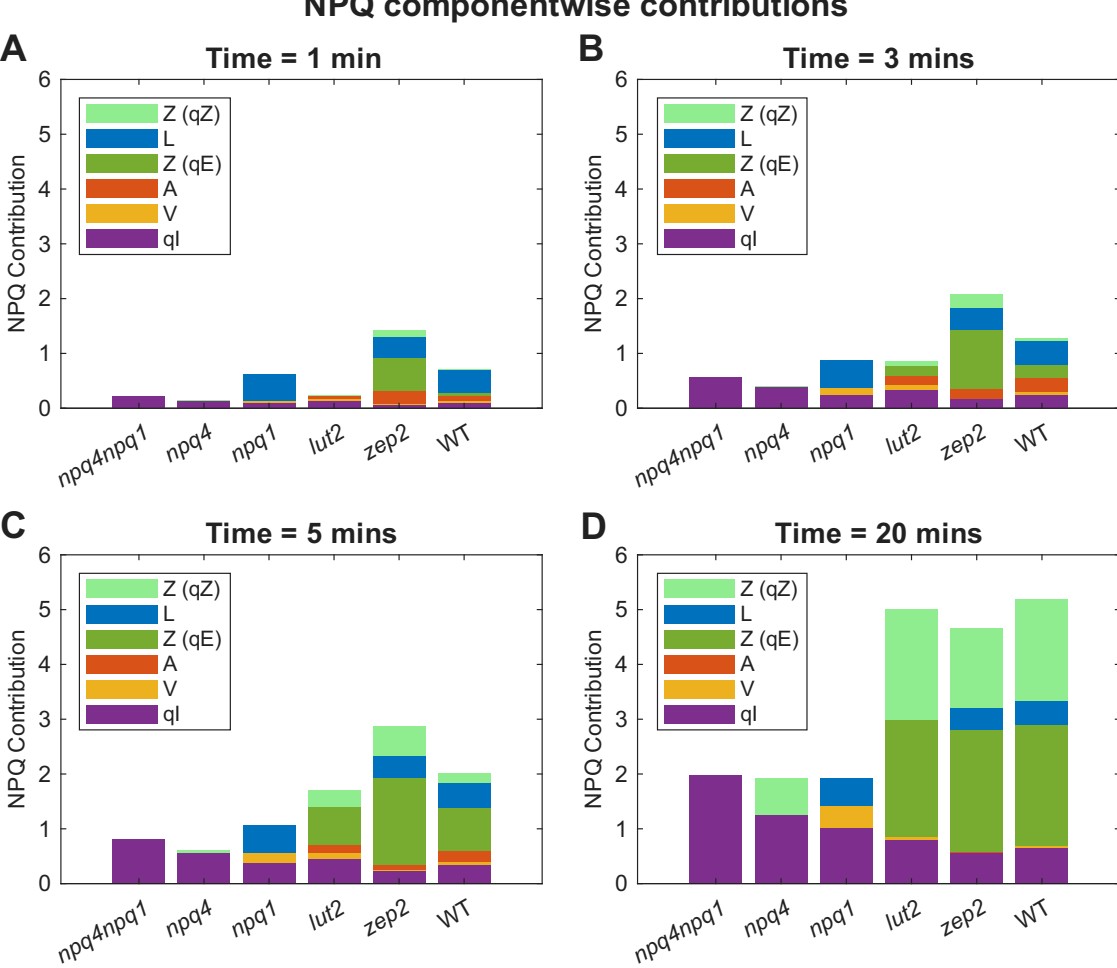

**Fig. 4 | The time dependent contributions of the components of NPQ in selected genotypes.** The NPQ contributions of qI, qE (V/A/Z/L), and qZ quenching components in *npq4npq1*, *npq4*, *npq1*, *lut2*, *zep2*, and WT under continuous high light at A t = 1 min, B t = 3 min, C t = 5 min, D t = 20 min. The NPQ contribution for each component was quantified by the linear relation $\kappa_{qX} \cdot [QX](t)$ for each quenching xanthophyll $QX$, $\kappa_{qZ} \cdot [Z](t)$ for qZ, and $\kappa_{qI} \cdot \alpha_{qI}(t)$ for qI. See Supplementary Movies 1 and 2 for the full time-resolved evolution of NPQ component contributions under 20HL and 5HL-10D-5HL, respectively.

absolute quenching contributions of Zea and Lut are similar on short timescales.

The quenching rate constants and per-molecule quenching effectiveness $Q_{eff}$ obtained from our model provide a quantitative basis for comparing the relative efficiencies of key NPQ processes across different xanthophylls. Formulating our modeled quenching rates in terms of quenching rate constants times concentrations ($\kappa_{QX}[X]$) allows us to quantify the differing quenching abilities of the key components of the photoprotective system in a time-dependent manner following exposure of the leaf to excess light (Fig. 4; Supplementary Movie 1). Based on model predictions, over the course of 20 min of HL exposure, photoinhibition increases in all genotypes, although those with strong NPQ capabilities show slower qI accumulation, as expected. Thus, although qI is not modeled mechanistically, we are still able to observe the effects of NPQ activity on the accumulation of photodamage. In WT, from $t = 0$ to $t = 3$ min (Fig. 4A, B), Lut and Ant are the major quenching contributors due to their rapid activation capabilities. Between t = 3 and t = 5 min, quenching contributions from Ant and Zea are similar in magnitude. After 5 min (Fig. 4C), Zea is the main quenching contributor. After 20 min (Fig. 4D), Zea is overwhelmingly dominant, suggesting that Zea (via both qZ and qE$_Z$) is the most important player in long-term defense against light stress. A similar pattern is observed in *lut2* and *zep2*,

albeit accelerated in *zep2* due to the pre-existing presence of Zea in the dark, and without Lut contributions in *lut2*. In *npq1*, the lack of the VDE-mediated VAZ cycle results in increased Vio contribution to quenching, and Lut engages rapidly in quenching at the onset of illumination, consistent with our observations in WT. In *npq4* and *npq1npq4*, which lack most major NPQ pathways, qI accumulates at an increased rate and acts as the major quenching contributor, explaining the lack of NPQ recovery in darkness. Additionally, Supplementary Movie 2, which shows the same NPQ component decomposition under the 5HL-10D-5HL actinic light sequence, visualizes how these components contribute to the NPQ relaxation in darkness and NPQ re-initiation upon the second HL period. Interestingly, NPQ contributed by Lut (qE$_L$) relaxes the fastest after the transition from HL to darkness, diminishing within ~1 min in WT, followed by qE$_Z$. By contrast, qZ decays much more slowly in darkness, aside from the slowest component, qI. Notably, *npq1* shows faster NPQ re-initiation upon the second HL period (see Results and Supplementary Fig. 4) despite the absence of Ant or Zea, supporting the rapid switching of qE$_L$ discussed above. These results highlight the importance of a time-resolved dynamic picture when interpreting the contributions of different quenching components, since they are highly time-dependent and strongly shaped by illumination duration and history.

# VPZ mutant $\tau_F$ predictions

**Fig. 5 | Model predictions for $\tau_F$ dynamics in logically constructed overexpression mutants under the 5HL-10D-5HL light sequence. A** VDE activity upregulated by 2X, 5X, and 25X from WT. **B** ZEP activity was upregulated by 2X, 5X, and 25X from WT. **C** PsbS activity was upregulated by 2X, 5X, and 25X from WT.

**D** Combined modeled upregulation of VDE, ZEP, and PsbS: VDE activity upregulated by 1X and 10X, ZEP activity upregulated by 5X and 50X, and PsbS activity upregulated by 2X and 5X. White and black bars indicate high-light (HL) and dark (D) phases of the actinic light sequence.

The complete WT model further provides a predictive framework for guiding systematic modifications to *N. benthamiana* aimed at improving NPQ induction and relaxation dynamics, with the ultimate goal of enhancing crop yields. In particular, overexpression of VDE, ZEP, and PsbS has been shown to increase biomass in certain species (*N. tabacum* and soybean/*G.max*), hypothetically by maintaining a WT level of quenching under HL while enabling faster relaxation in low light or darkness[67,68]. However, similar enhancements were not observed in *A. thaliana* and potato/*S. tuberosum*[69,70]. While the reasons for these species-specific discrepancies remain unclear[71], some studies suggest that optimizing the stoichiometries of these three proteins is crucial, depending on their native qE and qZ capacities, to achieve both faster, but not greater, quenching under HL and, more importantly, enhanced recovery in low light or darkness[68,71,72]. Building on prior empirical studies, our model enables predictive simulation of NPQ behavior under different expression levels and stoichiometries of these key enzymes. VDE and ZEP overexpression is modeled by increasing $k_{va/az}$ and $k_{za/av}$, respectively. PsbS overexpression is modeled by increasing the PsbS-dependent rate $k_{QX}$ by the same factor across all xanthophylls. Model predictions (Fig. 5) suggest that inducing a 2:10:1 expression ratio of VDE to ZEP to PsbS enhances both NPQ activation and relaxation rates and amplitudes. Additionally, overexpressing ZEP and PsbS by five- and two-fold, respectively, results in NPQ behavior considered optimal for improving crop yield—achieving faster but equal quenching in HL while allowing greater recovery in

darkness compared to WT[67]. In these modeled VPZ overexpression mutants, an increase in the relative contribution from qE$_V$, qE$_A$, qE$_Z$, and qE$_L$ underlies the improvements in the NPQ induction and recovery dynamics. While this thought experiment offers helpful insight, future experimental work with VPZ mutants of *N. benthamiana* or related species will be necessary to validate these predictions and help realize their potential for field application.

While the kinetic model presented here successfully captures NPQ dynamics in *N. benthamiana*, several limitations must be considered. Biological variation introduces uncertainty in parameter estimation, as differences in growth conditions and leaf age affect pigment pools, enzyme activity, and consequently, fluorescence lifetime and xanthophyll interconversion rates—even within the same genotype. These factors likely contribute to the model's slight NPQ underestimation of WT NPQ under the 3HL-1D-1HL-3D-9HL-3D sequence, as mentioned in the Results section ("Predictions made by the NPQ model"). While biological replicate averaging and normalization in this model (using $\tau_F(0)$ as an initialization parameter) help mitigate variability, fully resolving it remains challenging. In addition, the model also simplifies the illumination treatments by using three designed regular and irregular HL-to-D illumination sequences. These were chosen to capture variation in NPQ activation and relaxation kinetics contributed by different components, with progressively more challenging conditions for model prediction, providing a controlled platform to interrogate NPQ mechanisms before extending the model to more complex light

regimes. However, natural sunlight exhibits more unpredictable and unpatterned HL-to-LL fluctuations[73], and more gradual transitions could influence both the relative contributions and the relaxation kinetics of different NPQ components. Additionally, the interplay between light intensity and NPQ induction—particularly the threshold-dependent activation of different molecular players—remains an open question[74]. Incorporating more stochastic illumination patterns, HL-LL transitions, and light intensity-dependent rate constants may improve the physiological relevance of this framework for understanding NPQ regulation under natural light fluctuations, if not offset by increased phenotypic variability. Lastly, to focus on the rapid NPQ components that occur in the PSII, our experimental approach involves closing all RCs using laser pulses in the region of the leaf from which Chl fluorescence is detected. In natural environments, however, RCs are likely to exist in both open and closed states, which may influence NPQ behavior. This situation was previously modeled using PAM fluorometry by Snellenburg et al.[44]. Recent work from our group has shown that NPQ reduces the range of excitation migration and limits the number of accessible RCs[57]. Thus, combining the approach of Snellenburg et al. with our current method would introduce a complex dependence on membrane morphology.

Despite these limitations, the model provides a valuable quantitative framework for dissecting different types of NPQ components in land plants. Its ability to resolve the time-dependent contributions of different molecular players and predict both fluorescence lifetime and xanthophyll dynamics suggests it can serve as a foundation for future refinements, improving predictions on the impact of key genetic modifications on NPQ dynamics and crop yields.

## Methods

### Plant material and growth conditions

Transgenic *Nicotiana benthamiana* (accession *Nb-1*) lines were generated via *Agrobacterium*-mediated transformation by the Ralph M. Parsons Foundation Plant Transformation Facility at UC Davis (https://ptf.ucdavis.edu/), as described in a previous work[26]. The newly characterized *zep2* mutant was generated using the same approach; the corresponding guide RNAs are listed in Supplementary Table 3, together with other previously reported mutants used in this study[26]. Higher-order quadruple mutants were generated by crossing homozygous, transgene-free double mutants (e.g., *npq4* or *psbs1 psbs2* crossed with *npq1* or *vde1 vde2*). Progeny were phenotypically screened and genotyped to identify homozygous mutants with the appropriate mutations, as described below. *N. benthamiana* plants were grown with a 10-hour daylength in a south-facing greenhouse. Seeds were germinated directly on a mixture of four parts Sunshine Mix #1 (Sungro) and one part perlite. Plants were fertilized with JR Peter's Blue 20-20-20 fertilizer monthly.

### Genotyping of CRISPR/Cas9 edits

Multiplexed CRISPR/Cas9 mutagenesis was employed to knockout *N. benthamiana* orthologs of key *A. thaliana* NPQ proteins. Orthologs of candidate NPQ genes (NPQ4/PSBS, NPQ1/VDE, NPQ2/ZEP, and LUT2) were identified using the *N. benthamiana* draft genome sequence v1.0.1 (Sol Genomics Network)[75] and *A. thaliana* protein sequences as queries. Gene structure was largely similar across pairs of genes, excluding LUT2-2, which was manually assembled by splicing two draft contigs (Supplementary Fig. 7A). Guide RNA (gRNA) spacers were cloned into a modified pCAMBIA2300 backbone for CRISPR/Cas9 mutagenesis, and an *A. thaliana* U6-26 promoter drives expression of candidate gRNAs. Candidate gRNAs were identified using CRISPR-P (crispr.hzau.edu.cn)[76] and two high-scoring candidates for each gene were chosen depending (1) on their ability to target both *N. benthamiana* orthologs and (2) sequence similarity to the orthologous *A. thaliana* gene downstream of the chloroplast transit peptide sequence. Each set of 2 gRNAs was synthesized as a gBlock (Integrated

DNA Technologies), interspersed with a gRNA scaffold and tRNA linker to allow for polycistronic gRNA expression as previously described[77] (Supplementary Fig. 7B).

Genomic DNA from putative knockout lines was isolated and genotyped by Phire Plant Direct PCR Master Mix (ThermoScientific™, Catalog #F160L) using the supplied dilution buffer. DNA was amplified by PCR using primers that spanned the two gRNA target sites for each gene of interest, with primer pairs specific to one of the two highly similar paralogs. In our previous study, we reported the gene models and gRNA target-site positions for the *N. benthamiana* NPQ genes used for mutagenesis in Supplementary Fig. S1[26], together with the corresponding gene annotations. Here, we present an updated gene model, including ZEP1 and ZEP2, and the corresponding gRNA target sites in Supplementary Fig. 7. Previously generated mutant lines and their knockout alleles are reported in Tables S1–S3 of our previous study[26]. PCR products were purified and sequenced by Sanger sequencing using the primers listed in Supplementary Table 4 (newly reported in this study) and, for previously generated genotypes, in Table S4 of our prior study[26].

The knockout alleles for single trait and higher order mutants (newly reported in this study) are listed in Supplementary Table 5 and Supplementary Table 6, respectively. Higher-order mutants were generated by crosses of homozygous single-trait mutants that lacked Cas9, which mitigated the concern for off-target mutations or the need for back-crossing. All four genes were genotyped for each higher-order mutant to confirm the homozygous knockouts as described.

Segregation of the Cas9 transgene was determined via changes in Chl fluorescence after antibiotic treatment as previously described[78]. Briefly, floated leaf punches were incubated in 50 μg/mL kanamycin (from a 50 mg/mL stock in Milli-Q $H_2O$) for 3-4 days under 100 μmol $m^{-2}$ $s^{-1}$ fluorescent light (14 h light/10 h dark; 27 °C day/25 °C night). Kanamycin sensitivity was quantified using an Imaging-PAM Maxi (Walz) by measuring Fv/Fm after 30 min dark acclimation in biological duplicates. Transgene-free (kanamycin-sensitive) plants showed a ~ 0.3–0.4 decrease in Fv/Fm, whereas transgenic plants maintained near-WT Fv/Fm (-0.6–0.8) across at least two replicates per plant screened (Supplementary Fig. 7C).

As mentioned in the Results section, ZEP is necessary for the production of Vio, Ant, and neoxanthin, the latter being an important precursor for the stomata-regulating plant hormone abscisic acid[61]. Correspondingly, knockouts of ZEP should constitutively accumulate Zea even in the dark. However, mutations in single and double mutants of ZEP also corresponded to significant changes in plant fitness and overall physiology. The *zep1zep2* mutant was only able to grow under the high-humidity conditions used for germination and was unable to maintain turgor pressure or grow past the seedling stage necessary for seed set (Supplementary Table 5). In contrast, both *zep1* and *zep2* mutants were able to grow under ambient greenhouse conditions with more modest differences in growth. While the *zep2* mutant was indistinguishable from WT plants, the *zep1* mutant exhibited stunted growth, frilled leaf edges, and reduced leaf turgor. Given the near-lethality of the *zep1zep2* double knockout, only one of eleven T0 transformants, ZEP-ko-4, produced viable, stable progeny (Supplementary Table 5). Raw Sanger sequencing chromatograms confirming edits in *zep1*, *zep2*, and *zep1zep2* lines (file identifiers correspond to progeny in Supplementary Table 5) are available at https://doi.org/10.5281/zenodo.16755870.

### High-performance liquid chromatography of whole-leaf extracted pigments

Leaves were harvested from *N. benthamiana* plants that had been dark-acclimated overnight and were cut into equal-sized disks (-0.5 cm). These leaf disks were exposed to dark or light conditions (-1500 μmol $m^{-2}$ $s^{-1}$) under a white-light LED panel. Each leaf disk was flash-frozen in liquid nitrogen, collected in tubes containing Lysing

Matrix D beads, and homogenized using a High-Speed Homogenizer (6.0 m/s, 1 × 40 s; FastPrep-24 5 G™; MP Biomedical). Pigments were extracted twice with 200 µL 100% ethanol until the remaining leaf debris, pelleted after 24,100 g centrifugation for 5 minutes, turned white to ensure complete extraction. To minimize pigment degradation and solvent evaporation, the extraction process was performed on ice or at 4 °C and in the dark. Approximately 400 µL of the pigment solution was collected from each sample, filtered through a 0.2 µm nylon filter, and concentrated to ~150 µL under nitrogen flow for 5 minutes before loading into an HPLC vial. The total leaf-pigment profile, including neoxanthin, violaxanthin, antheraxanthin, lutein, chlorophyll-*b*, zeaxanthin, chlorophyll-*a*, and β-carotene, was obtained by performing high-performance liquid chromatography (1100 HPLC, Agilent) with a combination of solvents consisting of methanol, methyl-*tert*-butyl ether, and water[79] for Supplementary Fig. 1 or a C30 column (YMC Carotenoid, 5 µm, YMC America) with a gradient of solvents consisting of methanol, methyl-*tert*-butyl ether, and water[80] for Supplementary Fig. 2. Pigments were quantified against standard dilution series.

## Whole-leaf fluorescence lifetime measurements and analysis

We used a lab-constructed time-correlated single photon counting (TCSPC) setup, similar to a previously described design[23,81], to measure Chl fluorescence lifetimes and quantify the quenching kinetics of the leaf samples. A diode laser (Coherent Verdi G10) operating at ~532 nm was used to pump a Ti:sapphire oscillator (Coherent, Mira900f, 76 MHz), generating pulses centered at ~808 nm. The laser beam was then directed into a beta-barium borate crystal, thereby frequency-doubled to ~404 nm to selectively excite the Chl *a* Q$_x$ band. A beam splitter directed a portion of the beam to a sync photodiode (Becker-Hickl, PHD-400) to provide time references to the TCSPC card (Becker-Hickl, PSC-150). The remaining beam, serving as the excitation source, was directed at the adaxial leaf surface at an angle of approximately 70°, carefully avoiding the leaf stem and petioles.

Prior to the measurements, plants were dark-acclimated overnight, and water was added to a custom leaf holder with a water reservoir to maintain leaf hydration throughout the entire measurement. The excitation beam was adjusted to 1.0 mW to saturate the RCs. The leaves were treated by one of the three 20-minute actinic light sequences using a Leica KL1500 LCD light source, with high light (HL) set at approximately 1500 µmol m$^{-2}$ s$^{-1}$. The sequences were designated as follows, with the numbers indicating the duration of each highlight (HL) or dark (D) period in minutes.

**20HL**: continuous high light for 20 minutes.

**5HL-10D-5HL**: 5 minutes of high light, 10 minutes of dark, followed by another 5 minutes of high light.

**3HL-1D-1HL-3D-9HL-3D**: a more complex sequence combining shorter intervals of high light and dark.

Chl *a* Q$_y$ band fluorescence emission was selected at 680 nm by a monochromator (HORIBA Jobin-Yvon; H-20) and detected by a microchannel plate (MCP)-photomultiplier tube (PMT) detector (Hamamatsu R3809U MCP-PMT. The system's instrument response function (IRF) had a full-width half-maximum (FWHM) of ~37 ps, sufficient to resolve sub-nanosecond fluorescence lifetimes. A LabVIEW program was applied to control a series of shutters and coordinate the excitation beam, actinic light, and detector according to the sequences.

Fluorescence decay profiles were recorded with a laser excitation and detector integration time of 1 second (termed a "snapshot"), collected every 15 seconds to ensure sufficient temporal resolution. A total of 83 snapshots were taken over any type of the 20-minute sequences, including 3 initial snapshots in the dark-acclimated state. From each 1-second integrated decay, a 0.2-second timestep with the longest lifetime was selected for analysis to further ensure that the RCs

were fully saturated[81]. Each fluorescence decay profile was fitted with a bi-exponential decay function, and the amplitude-weighted average lifetime was calculated as follows:

$$\tau = \frac{\sum_i A_i \tau_i}{\sum_i A_i} \tag{5}$$

Here, A$_i$ and τ$_i$ denote the amplitudes and fluorescence lifetimes of the i$^{th}$ fitting component, respectively. The fitting accounts for the convolution of the fluorescence signal with the instrument response function (IRF), using the following equation:

$$I(t) = \int IRF(t') \sum_i A_i e^{-\frac{(t-t')}{\tau_i}} dt' \tag{6}$$

where $I(t)$ is the measured fluorescence intensity, $IRF(t')$ is the instrument response function, and A$_i$ and τ$_i$ are as defined above.

Note that the measured Chl fluorescence decay results from the averaging of tens of thousands of rate constants. As we showed in previous work[82], a decay curve arising from 30,000 rates can be perfectly fit by a sum of two exponentials, though these components do not correspond to any specific underlying rates or combinations thereof. The amplitude-weighted average lifetime values obtained from each snapshot were directly used in constructing and testing the NPQ model. Each lifetime thus corresponds to a single data point in the figures. A representative example of the raw fluorescence decay profile, bi-exponential fit, and IRF is shown in Supplementary Fig. 8.

## Modeling details

NPQ dynamics are modeled by a set of biochemical pathways describing the VAZ cycle and a light-dependent quencher activation mechanism. Each interconversion between species shown in the NPQ model schematic, as shown in the Results, is treated as an elementary reaction step described within a set of differential equations. Rate kinetics are determined by rate constants and initial concentrations (mmol/mol Chl *a*) of violaxanthin and membrane protein binding sites, which are parameters fit to experimental data.

The fit quality of a parameter set, $\delta^2(\theta)$, is quantified by performing a least-squares residual fit to experimental lifetime data for a given D-L sequence S:

$$\delta^2(\theta) = \sum_{i=1}^{N_{snaps}} \left( \tau_{\exp}(t_i, S_i) - \tau_{model}(t_i, S_i, \theta) \right)^2 \tag{7}$$

For a given parameter set, the kinetic differential equation system is solved with MATLAB ode23t and $\tau_F(t)$ and $\delta^2(\theta)$ are calculated directly from the solution set. *particleswarm* from the MATLAB Global Optimization Toolbox is used to perform particle swarm optimization searching parameter space for fit quality minima. The output Θ is refined using *patternsearch* and *fmincon*. 95% confidence intervals were obtained for the final parameter set via refitting to experimental data and bootstrapping to estimate lower and upper bounds.

Full details of the kinetic scheme and the complete parameter set with 95% confidence intervals are given in the Supplementary Information (Supplementary Methods).

## Reporting summary

Further information on research design is available in the Nature Portfolio Reporting Summary linked to this article.

## Data availability

The data supporting the findings of this study are available within the article and at https://doi.org/10.5281/zenodo.16755870.

## Code availability

The codes for NPQ models used in this study are available at https://doi.org/10.5281/zenodo.16755870.

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

## Acknowledgements

We thank Dr. Thomas P. Fay for kindly providing codes from the previous studies and insight on modeling based on the xanthophyll cycle. We thank Dr. Masakazu Iwai for his insights on HPLC measurements. This work was supported by the U.S. Department of Energy, Office of Science, through the Photosynthetic Systems program in the Office of Basic Energy Sciences (K.K.N. and G.R.F.). R.L. was supported in part by the U.S. Department of Energy, Office of Science, Office of Workforce Development for Teachers and Scientists (WDTS) under the Science Undergraduate Laboratory Internships (SULI) program. D.P.T. was supported by the Berkeley Fellowship and the NSF Graduate Research Fellowship Program (Grant DGE 1752814). K.K.N. is an investigator of the Howard Hughes Medical Institute.

## Author contributions

L.L. and R.L. contributed equally. G.R.F. and L.L. conceived the research. D.P.-T. and S.A.M. generated and screened for *N. benthamiana* NPQ mutants. L.L. conducted the TCSPC experiments and initial data analysis, with technical assistance from H.E.L., A.M., and R.L. L.L. performed HPLC experiments and analysis with input from D.P.-T. and H.H. R.L., and T.-Y.L. developed the model, with R.L. implementing it and performing the final data analysis. L.L., R.L., and G.R.F. drafted the manuscript with input from D. P.-T., K.K.N., and T.-Y.L.

## Competing interests

The authors declare no competing interests.
