## [Transparent Peer Review file · Nature Communications]

Dissecting the Contributions to Non-photochemical Quenching in a Land Plant Under Fluctuating Light

Corresponding Author: Professor Graham Fleming

A version of this paper was originally rejected for publication by Nature Communications, however that decision was reconsidered after appeal by the authors.

Version 1:

Reviewer comments:

Reviewer #1

(Remarks to the Author)

This report presents an extensive body of work concerning the generation of various tobacco mutants, each deficient in specific proteins associated with non-photochemical chlorophyll fluorescence quenching (NPQ). The authors undertook chlorophyll fluorescence lifetime measurements across all mutant lines, charting both the induction and relaxation kinetics of NPQ. They proceeded to construct a kinetic model designed to incorporate the various components—hypothetically corresponding to discrete phases of NPQ—with the intention of elucidating the respective roles of xanthophylls (notably lutein and zeaxanthin) and the PsbS protein in the plant's response to fluctuating light conditions. While the experimental ambition is commendable, one cannot help but observe that similar Arabidopsis mutants have already been extensively developed and scrutinised in the past. Regrettably, this current study appears to overlook a significant corpus of prior findings. Notably, there remains no compelling *in vivo* evidence that zeaxanthin or lutein are directly responsible for quenching chlorophyll fluorescence. The mechanism attributed to the so-called qZ component of NPQ is, as yet, not fully resolved. Indeed, several reports suggest that the observed quenching is more consistent with qI-type behaviour—dependent upon residual proton accumulation in the PSII antenna system, rather than any direct engagement by carotenoids. Furthermore, the qI component itself remains a complex and multifaceted phenomenon. Unless it can be unequivocally attributed to irreversible damage to reaction centres (RCII), it must be considered a conflation of both photoinhibitory damage and regulatory quenching within the antenna. More broadly, the manuscript presents a somewhat partial narrative, both in interpretation and citation. Several pivotal contributions to the mechanistic understanding of NPQ, most notably the kinetic analyses and spectroscopic investigations of Horton, Holzwarth, and Jahns, are conspicuously absent from the discussion. The model fitting, in several instances, appears inadequate, and the exclusion of alternative kinetic frameworks (such as those developed by van Grondelle and colleagues) is a notable omission. Finally, one is left to question the practical utility of the modelling approach adopted. Its level of formalism, while impressive, does not appear to yield predictive power beyond that already demonstrated in seminal work such as the Science article by Kromdijk et al. (Long and Niyogi), wherein overexpression of PsbS, VDE, and ZEP was shown to confer tangible physiological benefit. In conclusion, while the effort is appreciable and the experimental execution technically competent, the manuscript suffers from conceptual oversights, selective engagement with the literature, and methodological opacity. A more balanced and integrative approach would be required for the work to reach its full scholarly potential.

(Remarks on code availability)

Reviewer #2

(Remarks to the Author)

The authors have developed a relatively simple model to estimate chlorophyll fluorescence lifetime based on the dynamics of quencher molecules and have successfully reproduced the activation and inactivation kinetics of NPQ under fluctuating light. Compared to the fluorescence intensity measurement by the PAM method, which has been commonly used to assess NPQ activity in plants, the new method provides a more accurate estimate of the concentration changes of multiple quencher molecules. The widespread use of this method will enable more accurate and precise phenotypic analysis of

mutant plants and is expected to contribute to plant breeding. The authors have shown that photosynthetic productivity might be improved by enhancing the three key enzyme activities of NPQ in specific ratios. Although the prediction has not been validated by creating overexpressing strains, the value of this paper is enhanced by presenting predictions that can be demonstrated or disproven by future research. I hope this manuscript will be published after minor revisions described below.

1. The explanation of the fluorescence lifetime model is dispersed into three sections: results, methods, and supplementary information, making it difficult to understand the entire model. I would like the supplemental materials to fully describe this methodology, including the portions already described in the Results and Methods.
2. The manuscript includes many kinds reaction rates and rate constants, but does not well explain what they mean. Even though it is possible to guess what the subscripts f, b, and eq mean, it is unkind that they are never explained in the text.
3. The schematic of stepwise fitting shown in Figure 1B does not seem accurate. lut2 and zep2 have different antenna compositions, so the qE-VSZ columns should be independent of each other, and lut2 should be marked with an * to clarify the difference.

(Remarks on code availability)

Reviewer #3

(Remarks to the Author)

The ms by Lam et al describes a modelling approach where different subcomponents of NPQ are separated from chlorophyll fluorescence lifetime data in wt and mutants of *Nicotiana*, grown under a few conditions. The kinetic modelling builds in part on previous work, including analyses of various mutants, and a set of double mutants are generated and included in the study. The claim of the authors is that they construct a “fluorescence lifetime-based quantitative kinetic model that disentangles individual photoprotection components and, when integrated additively, accurately predicts wild-type and mutant quenching/recovery behaviors under various light-dark regimes”

I have several problems with the ms. I am not a modeller and may not understand all details well enough but from what I can see the approach rely on assumptions that may not necessarily be true, at least where there is far from a consensus on the matter. The NPQ field is hard to navigate in and very entrenched with fundamental disagreements between the key persons in the field concerning the photophysical mechanism of the quenching reaction. Some, like these authors, believe the quenching reaction is a direct quenching of the excited state of chl to the ground state mediated by a carotenoid. Others have different views, there exist also a lot of data in the literature that qE occurs through an interaction between LHCII and PsbS and that some kind of protein conformational change leads to quenching through chl-chl interactions. That camp tends to believe that zeaxanthin, and perhaps other carotenoids, in a yet unclear and perhaps more indirect way affect this change. The jury is still out on this subject but the authors do not even mention that their assumption is not accepted by many other experts in the field. Since the whole ms rely on this assumption, the fact that it is not consensus cannot simply be ignored. I myself do not know what to believe in – maybe the view of the authors is true – but a ms on the subject in a highly respected journal should not without comments/discussion rely on assumptions that many others think are wrong.

In addition:

- Related to what written above: Since only a direct quenching to the electronic ground state is considered, and a “Stern-Volmer” analog description is used to describe it, doesn't the design of this study simply exclude any other possible quenching models? An interaction between the excited state of the quenched Chl-protein complex and the quencher molecule could give rise to a new fluorescing species with different emission spectrum and lifetime. Fluorescence is detected at only wavelength (680 nm) and the kinetic analysis only rely on average lifetimes, wouldn't such a species be invisible in this analysis?
- Also concerning quenching mechanisms: In addition to the ones discussed and analysed here, qH was identified in the lab of the authors. It has also been discussed that spillover from PSII to PSI may occur (for example in <https://academic.oup.com/pcp/article/64/8/858/7148152>, <https://www.biorxiv.org/content/10.1101/2024.04.18.590023v2> and <https://www.biorxiv.org/content/10.1101/2025.01.26.634902v1>). Cyclic electron flow may also influence NPQ. Do the authors believe that these processes are irrelevant for their analysis? If so, motivate.
- Is there evidence that all these postulated protein PX complexes are formed in the membrane, and that the action mechanism for at least some of the carotenoid is only indirect in the membrane, without forming such PX complexes?
- More than 45 parameters - quenching efficiencies for various carotenoids, interconversion rates etc. - are fitted in the model. Doesn't this mean a severe risk of overfitting given the amount of independent experimental data? And what does this mean for the error range of the parameters? The number are sometimes given in tables with five digits, which cannot be appropriate. These aspects should be discussed.
- A 1 mW beam of laser pulsed measuring light is used. I assume that the same leaf spot has been exposed for ca 20 minutes. If I understand it correctly, this measuring light is much higher than the actinic light which would explain why even

the npq1/npq4 mutant has quite substantial quenching (ca 0.7). It is likely that in wt and other mutants, the measuring light may cause even bigger effects. I think the authors need to discuss this issue.

- The model performs well for WT under fluctuating light but less well under the 3HL-1D-1HL-3D-9HL-3D regime, or for example in zep2. What is the explanation for this?

- The model assumes a halved number of pigment binding sites in the lut2 mutant to approximate LHC structural changes. However, alterations in VAZ binding affinities and LHCII trimer stability may occur in the absence of lutein may have nonlinear effects on NPQ kinetics

- The model also assumes that quenching contributions are additive across mechanisms which may not necessarily be true. I guess this could be tested directly by comparing the modelled sum of two single mutant behaviours with the observed double mutant behaviour. Discrepancies here could reveal pleiotropic interactions or structural compensation that the model does not account for. Without this validation, the assumption of linear additivity remains a speculation.

- The authors talk about a "lifetime-based quantitative kinetic model that disentangles individual photoprotection components" and lifetime snapshots have been taken. But as far as I understand in their model they only use average lifetimes, not the multi-exponential kinetics. Is it then really a lifetime-based quantitative kinetic model? As lifetimes are proportional to the fluorescence yield, wouldn't it be possible to reach the same or similar conclusions from steady-state fluorescence measurements, just that the calculated rates in that case would be proportional and not absolute? If so, this analysis does really not make use of the extra information obtained from ultrafast time-resolved experiments.

The same lab has previously modelled the photoprotection in *Nannochloropsis oceanica*. In the introduction of this ms they lists a number of "significant limitations" of that model due to various assumptions. Although I acknowledge the skills of the lab and the complexity of this analysis, I am worried that some of the assumptions they have made here, and the issues mentioned above, reduces the value of this ms and the validity of the claims. Maybe the authors can present good explanations and good discussion on these points, but I am quite sceptic.

There are many papers on NPQ mechanisms which suffer from the same weakness; making assumptions that others do not buy. It is not easy when there is no consensus in the field about general assumptions but to move forward it is important to interpret data in the light of the whole body of literature, and this is where I think this ms fall short. Therefore I cannot recommend publication, at least not in the current form.

(Remarks on code availability)

Reviewer #4

(Remarks to the Author)

In this submission, the authors attempt to use a fluorescence lifetime modelling approach to understand the contributions of different components of NPQ in wildtype and mutant *Nicotiana benthamiana* plants. This review is written from the perspective of someone who has spent a lot of time measuring NPQ in whole plants but using PAM-style approaches and with little experience of modelling. I am therefore reading this as an interested reader rather than an expert and my review is delivered on this basis. I assume more expert reviewers have also been consulted.

The paper opens by mentioning the existence of 5 mechanisms of NPQ, though I was a little unclear from the introduction what they consider these to be. I assume that they are identifying Lut quenching as a distinct NPQ mechanism, or are they including this in qE, not quite clear from the wording. Also, I am not clear which are the canonical and non-canonical mechanisms. And is reaction center quenching, associated with Ca²⁺ release, included in your count or canon? Anyway, they then go on to develop a model that only considers 3 mechanisms of quenching. This could perhaps be more clearly explained in the introduction.

The main experimental approach involves the measurements of the decay kinetics of fluorescence under different conditions. I was a little unclear about the details of how these measurements were made, but I assume this is measuring the fluorescence decay following application of 404 nm flashes (duration?) and these decays were fitted with a bi-exponential decay function. It is not clear how good the fit is across different plants and conditions, as no examples are shown. The overall strategy of the modelling made sense to me – mutants with varying lesions in NPQ were fitted using their model and the parameters estimated were used predictively to compare to wild type and further mutants including in additional light regimes. It is always a concern with modelling that it is always possible to find a model to fit data, but using this in a predictive way gives confidence. The initial fits and predictions are not perfect but seem to capture the main processes occurring. Given the number of simplifications required this is reasonably convincing.

In terms of applying the models to predict the optimal solution for optimising NPQ, I was not totally sold on the usefulness of the approach, though I think this is a useful thought experiment. In the discussion, the authors are honest about the limitations of their approach, especially in terms of the need to consider biological variability. Whether this experimental approach will ever be as accessible as traditional PAM style measurements I do not know, but this does have the potential to provide useful information in at least some circumstances.

(Remarks on code availability)

Reviewer #5

(Remarks to the Author)

In this study, Lam et al. explore the specific contributions of NPQ components in *Nicotiana benthamiana* under fluctuating light conditions. By integrating chlorophyll fluorescence lifetime measurements on intact leaves with HPLC pigment profiling, the authors construct a kinetic model that quantifies the activity and molecular efficiency of distinct mechanisms (qE , qZ , qI) and of individual xanthophylls (Zea, Lut, Ant). Through the use of single and double mutants affecting specific NPQ pathways (some pre-existing and others generated in this study), they deconvolute overlapping processes and fit the model through parameter optimization based on experimental fluorescence decay data. The model computes the contributions of each quencher to energy dissipation over time and accurately predicts NPQ dynamics under complex light regimes. This represents an important methodological advancement, offering a predictive framework to guide genetic strategies aimed at improving light use efficiency and photoprotection.

Nevertheless, many biological complexities are simplified. The fluctuating light conditions were selected a priori, and the parameter values used in the model may be influenced by various factors. For example, in natural environments characterized by unpredictable sunflecks and alternating low/high light conditions, the model could potentially underestimate or overestimate NPQ dynamics. Or even the approximations used for qI . In fact, the authors themselves acknowledge the main limitations (lines 362 to 376). Nonetheless, the model is coherent with experimental data and provides a solid foundation for further refinement.

In the natural environment, the sudden changes in light intensity are accompanied by strong variations in the spectra of actinic light. A short paragraph could be dedicated to this factor of complexity and its effect on measurements.

Additional points:

- A sensitivity analysis of kinetic parameters (k) across NPQ components would improve robustness and highlight critical points to be implemented to cope with large biological variability.

- In Fig. S3, quenching induced by the laser in the absence of actinic light reaches ~ 0.6 ns after 20 minutes, which is non-negligible. How frequently are lifetime snapshots taken per induction curve? Reducing snapshot density might minimize laser pulse-induced quenching, especially during dark phases, and improve the accuracy of recovery kinetics; a potential trade-off.

- Line 334: please verify consistency with respect to Fig. 4A. The major components are Lut and Ant, not Vio.

(Remarks on code availability)

Version 2:

Reviewer comments:

Reviewer #1

(Remarks to the Author)

The authors replied to my comment point-by-point:

1. Choice of mutants

The authors argue that thylakoids are more suitable for transient absorption (TA) measurements, and I am prepared to accept that point in principle. However, the material has not been properly characterized with respect to electron transport and the resulting proton gradient (ΔpH) that drives NPQ. Both the kinetics and amplitude of these processes should be reported. Without such data, neither the measurements nor the resulting model can be considered reliable. Furthermore, reference 26 does not, in my view, provide higher-quality data than comparable work in *Arabidopsis*, nor does it present convincing evidence that zeaxanthin directly receives excitation energy from chlorophyll.

2. Zeaxanthin as a quencher

The authors appear unwilling to engage with the full scope of research on the relationship between zeaxanthin and NPQ published over the past few decades. Instead, they continue to promote the long-standing and now outdated view originating with Demmig-Adams et al. some 35 years ago. This selective treatment of the literature risks perpetuating a simplified narrative that does not reflect current understanding.

3. qI component

The model presented does not question the mechanistic origins of qI but rather addresses only its kinetic behavior. This raises the question: what is the real utility of such a model for physiologists? Without a mechanistic framework, a purely kinetic description of such a complex component offers limited biological insight.

4. Kinetic aspects of NPQ

Even within the narrow kinetic focus, the treatment of NPQ dynamics in this manuscript remains incomplete. The relevant kinetic aspects of NPQ have not been adequately reviewed, and the analysis feels partial at best.

5. Suitability for Nature Communications

From the discussion surrounding the modelling work of Snellenburg et al., it is evident that this project is highly specialized and riddled with unresolved controversies. As such, it seems more appropriate for a specialist journal in photosynthesis research, rather than a broad-audience outlet such as Nature Communications. The current scope and accessibility do not justify publication in a general high-impact venue.

6. Authors' own admissions

The authors themselves acknowledge that their approach requires further experimental validation and, in all likelihood, improved quality in the kinetic fitting of their data. This further reinforces my view that the manuscript, in its current form, is not ready for publication in its intended journal.

(Remarks on code availability)

n/a

Reviewer #2

(Remarks to the Author)

Photoprotection mechanisms collectively referred to as non-photochemical quenching (NPQ) remain incompletely understood due to the large number of underlying components and their diversity among species. In this study, the authors established a convenient technique to monitor chlorophyll fluorescence lifetime changes accompanying the induction and relaxation of NPQ under fluctuating light conditions. Furthermore, they proposed a simplified NPQ model, composed of only the essential elements, that accurately reproduces these fluorescence lifetime dynamics. I recommend the prompt publication of this manuscript.

Compared with the commonly used PAM-based parameter, $(F_m - F_m')/F_m'$, the newly proposed $\tau F(t)$ has two distinct advantages: it is less influenced by residual NPQ present before the onset of measurements, and it exhibits higher linearity with respect to changes in quencher concentration. The authors applied model fitting to fluorescence lifetime data obtained from both wild-type plants and multiple NPQ mutants. Beginning with a double mutant containing the fewest NPQ components and progressively incorporating additional factors, they ultimately determined the parameter set representing all NPQ factors in wild-type plants. The validity of this NPQ model was confirmed by its ability to predict, with high accuracy, the fluorescence lifetime dynamics of independent NPQ mutants under fluctuating light conditions. The authors further suggest that this model could provide insights into improving photosynthetic responses and potentially contribute to crop yield enhancement.

In constructing the model, only essential components were deliberately included, ensuring neither redundancy nor omission. Consequently, the model does not aim to fully replicate the complex behavior of NPQ in natural environments, nor does it present novel discoveries regarding NPQ mechanisms. The authors explicitly state that these were not the primary objectives of their work.

Since the inception of NPQ research over half a century ago, non-invasive PAM measurements have greatly advanced the field. The approach proposed in this manuscript—combining fluorescence lifetime measurements with a streamlined NPQ model—offers the potential to detect subtle alterations in minor NPQ components that PAM-based analyses cannot resolve. This advance is expected to be valuable for future photosynthesis research and for practical applications such as crop breeding. The publication and subsequent broad validation of this method across diverse environments and species may eventually establish it as an alternative to conventional PAM-based techniques.

(Remarks on code availability)

Reviewer #3

(Remarks to the Author)

The ms has been modified in several ways to satisfy the concerns of me and the other reviewers. I think that the authors have done a good job in improving the explanation of what they have done and why. In particular, I appreciate the list of assumptions made and not made in the Extended methods section. But as not everyone immediately reads the SI, there should a specific reference to this information in the main text, there may be others that like me need this description.

Also, the references <https://academic.oup.com/pcp/article/64/8/858/7148152>, <https://www.biorxiv.org/content/10.1101/2024.04.18.590023v2> and <https://www.biorxiv.org/content/10.1101/2025.01.26.634902v1> I mentioned talk about PSI as a quencher of PSII, not state transitions (as mentioned in the rebuttal letter). Although it is still not well documented if this is a mechanism of importance in angiosperms including *Nicotiana* (and there is no useful mutant), what would it mean for the conclusions if spillover is there?

Anyway, I am still somewhat doubtful about the value of this contribution. Maybe I do not understand enough mathematics/biophysics to appreciate why this is suitable for Nature Communications, I feel that I do not really know more about NPQ after reading and thinking of the ms than I did before. If the other reviewers and the editor think differently I will not complain if it gets accepted, but I am not convinced that it is important enough...

(Remarks on code availability)

Reviewer #4

(Remarks to the Author)

The authors have engaged with my previous review in a constructive manner and addressed all the points I raised. As noted by the authors and raised by other reviewers, NPQ is an area in which there has been a lot of disagreement and argument over details. It is refreshing to see the approach here that attempts to take a broader view. The changes made by the authors improve, for me at least, the readability of the paper.

As noted in the previous review, I have only low expertise in the modelling aspects of the work, but assuming other referees are satisfied with this, I am happy to support publication.

(Remarks on code availability)

This aspect is outside my area of expertise.

Reviewer #5

(Remarks to the Author)

This is a revised manuscript that I evaluated already. Revisions took in due account criticisms and suggestions. Thus, the revised manuscript is presently suitable for publication and will constitute a valid tool for screening mutants for improved light use efficiency in different conditions. This is a valuable contribution for both basic and applied science. Having said so, Authors might have considered the results they have obtained with the (new) genotype ZEP npq4 of minor importance with respect to the general importance of the ms which deals with modelling. Nevertheless, the reduced lifetime in the dark in the ZEP npq4 points to an autonomous activity of Zea in quenching that could well be highlighted in the discussion.

(Remarks on code availability)

I did not review the code. But a co-workek in my lab has donloded it and found it usable. It will take some time for a more thoughtfull evaluation.

Version 3:

Reviewer comments:

Reviewer #1

(Remarks to the Author)

I appreciate the authors' detailed and thoughtful responses and acknowledge that the manuscript has been improved in several important respects, particularly with regard to clarification of modelling assumptions, expansion of the literature review, and more careful positioning of zeaxanthin's role in NPQ. However, despite these improvements, my central concerns remain unresolved, and I do not feel able to recommend publication of the manuscript in Nature Communications. Most importantly, the study still lacks experimental characterisation of electron transport activity and ΔpH generation in the specific thylakoid preparations used for the transient absorption and lifetime measurements. The authors rely on general literature and modelling assumptions to justify treating ΔpH as an instantaneous, constant-amplitude switch, but no direct or proxy measurements (proton gradient indicators or electron transport rates) are provided. Without such information, it is not possible to assess whether the kinetic parameters extracted from the model are biologically meaningful or comparable across genotypes. This limitation undermines confidence in both the experimental interpretation and the quantitative modelling. While the authors correctly state that their model is intended to be mechanistically agnostic and primarily kinetic, this does not remove the need for adequate physiological constraints on the system being modelled. In my view, the absence of ΔpH and electron transport characterisation remains a fundamental weakness, rather than a secondary or stylistic issue. The authors have also clarified their position regarding zeaxanthin and expanded their discussion of alternative models of NPQ, which is welcome. Nevertheless, the work still engages with long-standing and unresolved controversies in photosynthesis research in a way that, in my opinion, limits its accessibility and general appeal to the broad readership of Nature Communications. The study will likely be of significant interest to specialists in NPQ modelling and ultrafast spectroscopy, but its current scope and experimental grounding do not, in my view, justify publication in a general high-impact journal.

In summary, although I recognise the technical sophistication of the modelling approach and the effort invested in revision, I remain unconvinced that the manuscript has reached the level of experimental completeness, robustness, and general relevance required for Nature Communications. I therefore recommend decline, while noting that the work may be more suitable for a specialist journal in photosynthesis or biophysical modelling following further experimental validation.

(Remarks on code availability)

Reviewer #3

(Remarks to the Author)

The ping-pong between authors and reviewers has improved the ms further, I have no specific comments on the presentation. I still think that this fits better in a specialized journal (and Reviewer 1 seems to agree) but if the decision will be to accept, OK with me.

(Remarks on code availability)

Reviewer #6

(Remarks to the Author)

In my opinion the assumptions of the model are appropriately justified. I think part of the problem lies in a misunderstanding between the reviewers and the authors. The reviewer writes: "Most importantly, the study still lacks experimental characterisation of electron transport activity and ΔpH generation in the specific thylakoid preparations used for the transient absorption and lifetime measurements." It seems the reviewer assumes the measurements are done on isolated thylakoids and that transient absorption has been used. This confusion arises from the introduction where this is written: "To that end, we selected *Nicotiana benthamiana* as our model system due to its compatibility with transient absorption spectroscopic measurements, which facilitate investigation of potential qE quenching mechanisms and assessment of exciton diffusion length in relation to NPQ levels in isolated thylakoids". However, in the manuscript only TCSPC measurements on whole leaves are reported, which is not mentioned in the introduction.

The electron transport activity and ΔpH generation would be a matter of concern in isolated systems, but are well documented in intact leaves. The reviewer is right that it will take some time to build the pH gradient, but the time-resolution in this study is not very high, so it should not be a limiting factor.

I have one point which I would like to have clarified.

The authors write:

"The memory effect explored by Matuszynska et al. is an intrinsic part of our model⁴³, as it was in our earlier algal model^{51,52,337}, and is unraveled in greater mechanistic detail³³⁸ here, demonstrating the importance of the Ant intermediate to NPQ memory during dark periods." This is about the observation that NPQ is induced faster upon a second illumination, which could be due to the presence of Ant or remaining Zea. However, in Figure S4 the faster NPQ induction is also observed for npq1, which cannot form Ant or Zea. The model still fits the data well, so there must be another explanation than remaining Ant/Zea.

(Remarks on code availability)

Response to Reviewers

For clarity, we have color-coded this response letter as follows:

Black: Reviewer comments

Blue: Our responses

Red: Direct citations of the manuscript and underlining relevant changes made in the revised manuscript. (Line numbers refer to the no tracked-changes version; page numbers are consistent across both versions.)

General Response

We thank all five reviewers for their careful reading of our manuscript. In reading the reviews it became clear that we did not adequately describe the motivation for our approach and specifically what we assume and what we don't assume. We agree with reviewer 3 that "The NPQ field is hard to navigate in and very entrenched with fundamental disagreements between the key persons in the field concerning the photophysical mechanism of the quenching reaction." This is why our model does not assume specific quenching *mechanisms* and makes what we believe to be the minimal number of assumptions necessary to create a closed self-consistent model. In addition, we were less than precise in our use of the word "mechanism". In what follows and in the revised manuscript we reserve "mechanism" to refer only to specific molecular processes such as energy, charge transfer or the induction of structural change allosterically. We refer to the phenomenologically defined components of NPQ (qE, qZ, qI etc.) as "types of NPQ", "components of NPQ", or "contributions to NPQ".

Specifically, what we assume is listed below followed by the list of assumptions we did not make. We have added these lists to the SI "*Extended methods*" section and summarized the key points when we introduce the modeling approach in the main text.

Assumptions Needed

1. NPQ in plants is mainly contributed by qE (VAZ, and lutein), qZ, and qI. In our study, qT, qH, and chloroplast movement have negligible effects.
2. Each NPQ contribution contributes independently to the Chl fluorescence quenching, and can be therefore modelled separately.
3. Quenching is modeled using a Stern-Volmer-like relation between fluorescence lifetime and the concentrations of quencher molecules (QV, QA, QZ, QL, qI). The Stern-Volmer relation has been the standard method of modeling fluorescence quenching since its introduction in 1919⁴⁵.
4. qZ depends on the concentration of Zea in a light-independent manner.
5. Lut is assumed to be always protein-bound when present.

6. The activation of protein-xanthophyll complexes (PX) to quenching complexes (QX) is needed for the qE type of quenching to be activated. While the total pool of xanthophyll-binding sites P is fixed and shared among Zea, Vio, Ant, and Lut, the formation/dissociation kinetic rates are unique to each xanthophyll (Zea, Vio, Ant, Lut).

[qI Model]

7. The qI component is modeled phenomenologically as irreversible photodamage to PSII reaction centers, accumulating over time with a rate proportional to excitation (i.e., inversely to lifetime).
8. The reverse/repair process is assumed to be negligible within our timescale.
9. Quenching observed during laser-only exposure is attributed to qI and is assumed to be sufficiently captured by fitting qI parameters in *npq4npq1* data.

[Data Validation]

10. Time-resolved fluorescence snapshots are assumed to sufficiently capture the dynamics of all the NPQ contributions, with average lifetimes representing total quenching at each time point. The average lifetime is directly proportional to the fluorescence quantum yield. We have shown elsewhere that fluorescence decays of a model thylakoid membrane⁵⁷ containing roughly 30,000 rates can be fit very well with two exponentials, but those two components do not relate to the underlying microscopic dynamics in any direct way.

[lut2 specifically]

11. Considering the structural changes in the absence of lutein, the number of protein-xanthophyll binding sites ([P]_tot) is assumed to be different between WT and the *lut2* mutant. [P]_tot in *lut2* was approximated by modeling to be ~50% of [P]_tot in WT.

Assumptions Not Needed

1. By fitting the lifetime values directly, we do not assume zero initial quenching in all the mutants as was done by Short et al.³⁷. This is important for mutants such as *zep2*.
2. Despite utilizing some widely accepted biochemical processes⁶ involved in induction of qE (including the VAZ cycle), the model does not make assumptions on the actual physical mechanism of how these quenching processes happen. The model never claims to resolve ultrafast excited-state dynamics. For example, the model doesn't assume direct quenching (by energy transfer/charge transfer) of Chl excited states by Car.
3. The model does not resolve or define the exact mechanistic basis of qZ or qI. It only differentiates them kinetically from qE based on mutant behavior and pigment profiles.
4. Calculating per-molecule quenching efficiencies does not reflect direct mechanistic interpretation. It reflects the general impact on NPQ/lifetime caused by each species.

[Other NPQ components]

5. While qT mutants were tested (e.g., *stn7* mutants) and found to have negligible influence on our system over the timescale we studied, we do not reject or adopt any specific quenching mechanism. We do not claim these (qT, qH, Chloroplast movement/thylakoid membrane structural adjustment) are irrelevant to NPQ in general.

[Fitting Interpretation]

6. The model does not assume that NPQ component contributions (e.g., qE, qZ, qI) are additive in general. However, when making predictions, the model does compute overall NPQ as a sum of individual quenching components (see Assumption 2 - each mechanism is independent). However, our approach of fitting simpler mutants and directly applying their rates to complex genotypes, such as *zep2* and WT, with great accuracy shows that NPQ components can be considered as additive.
7. The model captures most of the NPQ dynamics in different illumination sequences. We designed different irregular sequences that presented a more difficult challenge for the model to make predictions.

The meaning of equations (1) and (2) in the main text and S1-S8 in the SI is, taking zeaxanthin as a specific example, that the extent of rapid NPQ (qE) depends on the amount of zeaxanthin bound to light-harvesting proteins and, for some components of NPQ, activation of quenching by PsbS. Beyond the proportionality of NPQ to the concentration of zeaxanthin, which is well established experimentally, and the requirement for activation of quenching via protonated PsbS, no further assumptions are made as to the molecular mechanism of zeaxanthin's role or the mechanism by which PsbS activates quenching other than requiring a ΔpH across the thylakoid membrane.

REVIEWER COMMENTS

Reviewer #1 (Remarks to the Author):

This report presents an extensive body of work concerning the generation of various tobacco mutants, each deficient in specific proteins associated with non-photochemical chlorophyll fluorescence quenching (NPQ). The authors undertook chlorophyll fluorescence lifetime measurements across all mutant lines, charting both the induction and relaxation kinetics of NPQ. They proceeded to construct a kinetic model designed to incorporate the various components—hypothetically corresponding to discrete phases of NPQ—with the intention of elucidating the respective roles of xanthophylls (notably lutein and zeaxanthin) and the PsbS protein in the plant's response to fluctuating light conditions.

Response: We thank the reviewer for their thoughtful and critical evaluation of our manuscript. We appreciate the recognition of the experimental scope and the effort to engage with the conceptual framework of our model. These comments have helped us identify areas where our

assumptions, scope, and citations could be more clearly articulated. Below we respond to each of their comments in detail.

1. While the experimental ambition is commendable, one cannot help but observe that similar *Arabidopsis* mutants have already been extensively developed and scrutinised in the past.

Response: We appreciate the reviewer's question and the opportunity to clarify our rationale for selecting *Nicotiana benthamiana* as the model organism in this study. Our choice was motivated by several practical and scientific considerations.

First, we note that fluorescence measurements do not allow detailed questions of molecular mechanisms to be fully addressed. Transient absorption experiments are required for this level of investigation. However, *Arabidopsis* thylakoids have proven difficult to use for these types of measurements. To the best of our knowledge, neither we nor others have successfully acquired high-quality, time-resolved transient absorption data from *Arabidopsis* thylakoid preparations. Also, the exceptional amount of biomass capable of being produced via *N. benthamiana* also dramatically improves biochemical experiment capacity that is often constrained by chloroplast and thylakoid yield. In contrast, we have obtained experimental evidence in isolated thylakoids for chlorophyll to zeaxanthin energy transfer²⁶ as well as intensity-cycling transient absorption data⁴² from five *N. benthamiana* NPQ mutants also described in this study. Although NPQ values are lower in thylakoids than in intact leaves, these measurements allow us to extract semi-quantitative estimates of photoprotection generated by a specific amount of fluorescence quenching. This makes *N. benthamiana* a uniquely suitable platform for linking fluorescence-based modeling with spectroscopic validation. We revised the manuscript shown below:

(Main text Page 6 Line 118-123) “To that end, we selected *Nicotiana benthamiana* as our model system due to its compatibility with transient absorption spectroscopic measurements, which facilitate investigation of potential qE quenching mechanisms²⁶ and assessment of exciton diffusion length in relation to NPQ levels in isolated thylakoids⁴²—measurements that are challenging to perform in *Arabidopsis thaliana*. This choice could enable both kinetic and mechanistic analysis of NPQ in land plants when combined with complementary techniques.”

Second, as mentioned in the Introduction, while our previous modeling efforts in *Nannochloropsis* offered important conceptual insights, that system lacks the complexity of higher plant NPQ regulation and is not representative of crop physiology. Developing the first land plant-specific NPQ model, even in a non-crop model organism, therefore fills a critical gap in the field.

Furthermore, the allotetraploid nature of *N. benthamiana* opens up new quantitative routes for titrating NPQ gene expression and downstream phenotypes. For example, we specifically highlighted our use of the *zep2* mutant (retaining functional ZEP1) in this study, and have refined the text to emphasize its novelty.

(Main text Page 8-9 Line 179-189) “In this study, we generated and characterized single and double mutants deficient in one or both ZEP orthologs. Vio is a precursor to abscisic acid, an essential plant hormone for stomatal regulation and drought tolerance. The lack of epoxidized Vio in the *zep1zep2* double knockout in *N. benthamiana* significantly reduced plant viability, with only one of eleven T0 transformants producing viable, stable progeny (see Methods). Each ortholog contributes additively to Zea epoxidation, with ZEP1 being the dominant copy. Interestingly, knockout of ZEP2 (hereafter *zep2*) resulted in a moderate knockdown phenotype, with WT-like growth and a near-native Vio and Neo composition and sufficient constitutive Zea to saturate Zea-dependent qE capacity (Fig. S1), unlike the non-native composition and slowed, humidity-dependent growth found in the *A. thaliana npq2* point mutant¹⁰. This *zep2* mutant provided quantitative and temporal variation in Zea-dependent qE and qZ that was necessary and sufficient for phenotypic modeling, without the pleiotropic impacts on xanthophyll composition that are present in existing *A. thaliana* mutants.”

The aforementioned intensity-cycling transient absorption study also utilizes a newly characterized *psbs1* mutant with reduced qE capacity, similarly deficient in one of two PsbS orthologs, which has expanded our capacity to interrogate the biophysical basis of NPQ. It is our hope that these mutants, parallel to those in Arabidopsis but with interesting use cases, can be a helpful resource to the NPQ research community.

Finally, as mentioned in the discussion, *N. benthamiana* shares close genetic similarity with *N. tabacum*, a species in which VPZ (VDE, PsbS, ZEP) overexpression has already been shown to improve biomass and photosynthetic performance⁵². We thus view *N. benthamiana* as an ideal model for bridging mechanistic understanding with translational relevance, particularly in the context of improving crop photoprotection through NPQ engineering.

2. Regrettably, this current study appears to overlook a significant corpus of prior findings. Notably, there remains no compelling in vivo evidence that zeaxanthin or lutein are directly responsible for quenching chlorophyll fluorescence.

Response: We thank the reviewer for raising this point. As noted in the general response, our model does not assume a specific direct quenching mechanism by zeaxanthin or lutein. While some studies support the possibility of energy or charge transfer from carotenoids^{24-26,30-35}, our model focuses on NPQ kinetics as well as the model’s predictive capability, and remains agnostic to the photophysical mechanism. We now clarify this distinction and the limits of our interpretation in the Introduction of the revised manuscript. There is a good deal of evidence (Demmig-Adams et al., Zeaxanthin Synthesis, Energy Dissipation, and Photoprotection of Photosystem II at Chilling Temperatures 1. *Plant Physiology* **90**, 894–898 (1989).) that the extent of qE is linearly related to the zeaxanthin content of leaves and this is what we assume.

(Main text Page 4 Line 74-86) “The specific molecular interactions and mechanisms involved in these NPQ processes are still extensively debated. In a recent review, Bassi and Dall’Osto⁶ describe four distinct mechanisms that involve either chlorophyll–

chlorophyll or chlorophyll–carotenoid interactions, where the role of the carotenoids may be direct^{24–26} or as an allosteric effector (e.g. in Chl–Chl interactions)^{27–29}. Other proposed mechanisms include charge transfer between Chl and Car that quenches Chl Qy^{30–33} and excitonic interactions that may produce a low-lying energy level of a Car, which relaxes rapidly to the ground state either directly or via transfer to a carotenoid S1 state^{34,35}.

Given the ongoing debate on molecular mechanisms (which may all occur in parallel), we have constructed a kinetic model for NPQ that is as agnostic as possible to the precise molecular mechanisms while incorporating the known factors of ΔpH activation of VDE and PsbS and the VAZ cycle. The time-dependent contributions of the various actors emerge naturally from the model and may help to advance the discussions of NPQ mechanisms.”

3. The mechanism attributed to the so-called qZ component of NPQ is, as yet, not fully resolved. Indeed, several reports suggest that the observed quenching is more consistent with qI-type behaviour—dependent upon residual proton accumulation in the PSII antenna system, rather than any direct engagement by carotenoids.

Furthermore, the qI component itself remains a complex and multifaceted phenomenon. Unless it can be unequivocally attributed to irreversible damage to reaction centres (RCII), it must be considered a conflation of both photoinhibitory damage and regulatory quenching within the antenna.

Response: We appreciate this important question. As noted, the model does not claim a mechanistic separation, but rather a kinetic distinction based on behavior in specific mutant backgrounds (e.g., *npq4*, *npq1npq4*).

In our model, qI is modeled phenomenologically by fitting *npq4npq1* data, representing a slow, irreversible process consistent with photoinhibition. While we do not attempt to resolve the underlying molecular nature of qI—acknowledging that it likely includes both damage-related and regulatory components—our goal was to quantitatively capture its kinetics under fluctuating light. We have added discussion to acknowledge this complexity and the limitations of our current treatment in our revised manuscript.

(Main text Page 10 Line 221-224) “This indicates although the laser baseline modestly contributes to detectable quenching, this effect is accounted for in our model within the qI parameters via the positive α_{qI} accumulation rate in darkness in *npq4npq1*, which was modeled phenomenologically due to the unresolved mechanistic basis of qI, yet still offers insight into the temporally resolved contribution of qI.”

4. More broadly, the manuscript presents a somewhat partial narrative, both in interpretation and citation. Several pivotal contributions to the mechanistic understanding of NPQ, most notably the

kinetic analyses and spectroscopic investigations of Horton, Holzwarth, and Jahns, are conspicuously absent from the discussion.

Response: We thank the reviewer for highlighting this oversight. In the revised manuscript, we have expanded our Introduction to include key foundational studies from these authors, please see Comment #2 of Reviewer #1. However, as we have noted, the extensive debate in the literature as to molecular mechanisms, while important, is not central to our model. Regarding the two sites, independent quenching mechanisms in each site introduced by Holzwarth and Jahns (Barbara Demmig-Adams et al., *Non-Photochemical Quenching and Energy Dissipation in Plants, Algae and Cyanobacteria*. (Springer Netherlands, Dordrecht, 2014).), we note that this is consistent with the additivity of contributions that our model suggests. We have added comments on this point in the Introduction of the revised text, also mentioned in our response to Comment #2 of Reviewer #1.

5. The model fitting, in several instances, appears inadequate, and the exclusion of alternative kinetic frameworks (such as those developed by van Grondelle and colleagues) is a notable omission.

Response: We understand the reviewer's concern for the quality of model fitting in some instances. While the model visually appears to underperform for certain mutants (e.g. *npq4zep2* in Fig 3G and Fig S6D and *lut2* in Fig S5C), we find the calculated RMSD values of fit quality ($\delta^2(\theta)$ defined in Methods) obtained for each mutant to be within an acceptable limit ($<1 \text{ s}^{-1}$). Furthermore, given that these discrepancies occur only in model predictions, rather than for sequences used for fitting, we anticipate and accept a certain level of error. Small deviations from model predictions are likely the result of pleiotropic interactions or specific mutant behaviors beyond the scope of our model.

We further thank the reviewer for highlighting past works on NPQ kinetic modeling by van Grondelle. We note that our work bears no contradiction to their previous studies. Their kinetic framework places emphasis on the state of PSII (as either “quenched” or “unquenched” and with “open” or “closed” reaction centers) without directly parameterizing any biochemical pathways or chemical species⁶⁰. Furthermore, in our experiments, we excite Chl *a* using a saturating laser pulse (1.0 mW) to ensure the reaction centers are closed during measurement; thus, in our context their model reduces to the interconversion of PSII between quenched and unquenched, which is effectively a maximal simplification of our model.

Nevertheless, their approach to modeling quantum yield (proportional to average lifetime) as a linear combination of fractional species concentrations and their quantum yields is intriguing. However, this approach restricts the characterization of species concentrations to fractions of a whole (constant) pool, which limits the ability to compare kinetics across different genotypes, which can exhibit large variations in the size of the total xanthophyll pool. As such, we consider such a framework inadequate for capturing the complex dynamics of NPQ in *N. benthamiana*.

(Main text Page 18-19 Line 418-425) “Lastly, to focus on the rapid NPQ components that occur in the PSII, our experimental approach involves closing all RCs using laser pulses in the region of the leaf from which chlorophyll fluorescence is detected. In natural environments, however, RCs are likely to exist in both open and closed states, which may influence NPQ behavior. This situation was previously modeled using PAM fluorometry by Snellenburg et al.⁶⁰. Recent work from our group has shown that NPQ reduces the range of excitation migration and limits the number of accessible RCs⁴². Thus, combining the approach of Snellenburg et al. with our current method would introduce a complex dependence on membrane morphology.”

6. Finally, one is left to question the practical utility of the modelling approach adopted. Its level of formalism, while impressive, does not appear to yield predictive power beyond that already demonstrated in seminal work such as the Science article by Kromdijk et al. (Long and Niyogi), wherein overexpression of PsbS, VDE, and ZEP was shown to confer tangible physiological benefit.

Response: We thank the reviewer for this question. The work by Kromdijk et al. demonstrated that overexpressing NPQ-related proteins (VDE, ZEP, PsbS) improved plant productivity, and included PAM-based NPQ measurements and representative pigment quantification⁵². However, it did not provide a predictive framework to simulate NPQ behavior under different light regimes or varying expression ratios. In contrast, our model enables quantitative simulation of NPQ kinetics across mutants and conditions, and makes testable predictions about how different levels of enzyme overexpression affect induction, relaxation, and overall quenching efficiency. These findings emphasize that the specific stoichiometry of overexpression is essential, and that stoichiometry is unknown for different species. This issue may reflect the differences in success in the application of the VPZ genetic engineering strategy across tobacco, Arabidopsis, potato, and soybean, reviewed in Croce et al.⁵⁶. We agree that further experimental validation is needed, and we now emphasize this in the revised Discussion section.

(Main text Page 17 Line 378-389) “The complete WT model further provides a predictive framework for ... Building on prior empirical studies, our model enables predictive simulation of NPQ behavior under different expression levels and stoichiometries of these key enzymes.”

(Main text Page 17 Line 395-399) “In these modeled VPZ overexpression mutants, an increase in the relative contribution from qE_v, qE_A, qE_Z, and qE_L underlie the improvements in the NPQ induction and recovery dynamics. While this thought experiment offers helpful insight, future experimental work with VPZ mutants of *N. benthamiana* or related species will be necessary to validate these predictions and help realize their potential for field application.”

Moreover, we emphasize that our model uniquely predicts the temporal evolution of quenching contributions from individual NPQ pathways, including qE_v, qE_A, qE_Z, qE_L, qZ, and qI,

throughout the course of light exposure (see Movie S1), a temporal resolution not currently achievable with existing studies.

7. In conclusion, while the effort is appreciable and the experimental execution technically competent, the manuscript suffers from conceptual oversights, selective engagement with the literature, and methodological opacity. A more balanced and integrative approach would be required for the work to reach its full scholarly potential.

Response: We acknowledge that our initial manuscript would have benefited from clearer articulation of the model's scope, assumptions, and relationship to existing literature, but most of the shortcomings noted by the reviewer relate to the misconception of how the model is constructed and what it is based upon. In response, we have made substantial revisions to clarify our modeling framework, explicitly distinguish between assumptions made and not made, and broaden the discussion to engage more fully with key mechanistic studies in the field. We hope that these revisions properly address the reviewer's concerns.

Reviewer #2 (Remarks to the Author):

The authors have developed a relatively simple model to estimate chlorophyll fluorescence lifetime based on the dynamics of quencher molecules and have successfully reproduced the activation and inactivation kinetics of NPQ under fluctuating light. Compared to the fluorescence intensity measurement by the PAM method, which has been commonly used to assess NPQ activity in plants, the new method provides a more accurate estimate of the concentration changes of multiple quencher molecules. The widespread use of this method will enable more accurate and precise phenotypic analysis of mutant plants and is expected to contribute to plant breeding. The authors have shown that photosynthetic productivity might be improved by enhancing the three key enzyme activities of NPQ in specific ratios. Although the prediction has not been validated by creating overexpressing strains, the value of this paper is enhanced by presenting predictions that can be demonstrated or disproven by future research.

I hope this manuscript will be published after minor revisions described below.

Response: We thank the reviewer for their positive and encouraging assessment of our manuscript. We are especially grateful for their recognition of the methodological contribution and the potential of our model to advance phenotypic analysis and predictive understanding of NPQ. We also appreciate the helpful suggestions for improving clarity and presentation. Below, we respond to each of the comments in turn.

1. The explanation of the fluorescence lifetime model is dispersed into three sections: results, methods, and supplementary information, making it difficult to understand the entire model. I would like the supplemental materials to fully describe this methodology, including the portions already described in the Results and Methods.

Thank you for this helpful suggestion. In response, we have reorganized and expanded the Supplementary Information “*Extended Methods*” section to provide a complete and self-contained description of the fluorescence lifetime-based kinetic model. This now includes all relevant equations, assumptions, and methodological explanations that were previously scattered across the main text and Methods. We hope this revision will improve accessibility for readers.

2. The manuscript includes many kinds reaction rates and rate constants, but does not well explain what they mean. Even though it is possible to guess what the subscripts f, b, and eq mean, it is unkind that they are never explained in the text.

Response: We appreciate this observation and apologize for the oversight. We have revised both the main text and Supplementary Information (Table S1 and “*Extended Methods*”) to clearly define all rate constants and subscripts upon their first use. We hope this improves clarity and readability.

(Main text Page 16 Line 350-351) “($\frac{k_{qxf}}{k_{qxf}+k_{qxb}}$, where k_{qxf}/k_{qxb} are the forward and backward rates of QX formation, respectively), and (3) the quenching rate constant of QX (κ_{QX}).”

(Figure S1) “Subscripts “f” and “b” denote forward and backward rates and superscripts “L” and “D” denote light and dark conditions, respectively. Additionally, subscripts “va” and “az” denote conversion from V to A and from A to Z, while subscripts “za” and “av” denote conversion from Z to A and A to V, respectively.

(SI “Extended Methods” Page 20) “To reduce the number of parameters needed to fit the model, we take VDE concentration as a ratio: $\alpha_{vde_a}(t) = \frac{[VDE_a]}{[VDE_{a,eq}^L]}$ where subscript “eq” denotes the equilibrium condition.”

3. The schematic of stepwise fitting shown in Figure 1B does not seem accurate. lut2 and zep2 have different antenna compositions, so the qE-VSZ columns should be independent of each other, and lut2 should be marked with an * to clarify the difference.

Response: We thank the reviewer for pointing this out. We have revised **Figure 1B** to more accurately reflect the structural differences between *lut2* and *zep2*, and now clearly mark with an asterisk (*) the qE-VAZ fitting component associated with *lut2* to indicate the mutant’s distinct xanthophyll-binding profile. Note that some of the rate parameters involved in the VAZ component of qE are shared between *lut2* and *zep2*, so in Figure 1B, the qE-VAZ section is not entirely independent between *lut2* and *zep2*, hence the overlap.

Reviewer #3 (Remarks to the Author):

The ms by Lam et al describes a modelling approach where different subcomponents of NPQ are separated from chlorophyll fluorescence lifetime data in wt and mutants of *Nicotiana*, grown under a few conditions.

Response: We appreciate the concerns regarding modeling assumptions, mechanistic interpretations, and broader context within the NPQ field. We address each point below in detail, with clarifications and revisions to the manuscript to reflect the reviewer's helpful suggestions.

We would like to clarify that all plants used in this study were grown under the same environmental conditions as described in the *Methods* section. However, they were exposed to different actinic light sequences during measurements in order to resolve the distinct kinetic behaviors of various NPQ components.

The kinetic modelling builds in part on previous work, including analyses of various mutants, and a set of double mutants are generated and included in the study. The claim of the authors is that they construct a “fluorescence lifetime-based quantitative kinetic model that disentangles individual photoprotection components and, when integrated additively, accurately predicts wild-type and mutant quenching/recovery behaviors under various light-dark regimes”

1. I have several problems with the ms. I am not a modeller and may not understand all details well enough but from what I can see the approach rely on assumptions that may not necessarily be true, at least where there is far from a consensus on the matter. The NPQ field is hard to navigate in and very entrenched with fundamental disagreements between the key persons in the field concerning the photophysical mechanism of the quenching reaction. Some, like these authors, believe the quenching reaction is a direct quenching of the excited state of chl to the ground state mediated by a carotenoid.

Response: As noted in the general response, our model does not assume or require a specific direct quenching mechanism between carotenoids and Chl*. While some experimental evidence supports such mechanisms (e.g., via excitation energy transfer or charge transfer), our model remains agnostic to the precise molecular mechanism of quenching. Rather than resolving ultrafast dynamics or structural details, our approach focuses on phenomenologically separating quenching contributions based on genotypic and pigment dynamics, rather than assuming or attributing them to a specific photophysical pathway. We revised the Introduction to make this clearer, please see response to Comment #2 of Reviewer #1.

2. Others have different views, there exist also a lot of data in the literature that qE occurs through an interaction between LHCI and PsbS and that some kind of protein conformational change leads to quenching through chl-chl interactions.

Response: We thank the reviewer for highlighting this important perspective. We agree that alternative mechanistic hypotheses for qE—such as PsbS-induced conformational changes or Chl-Chl interactions—are widely discussed in the field. We have revised the manuscript, shown in our response to Comment #2 of Reviewer #1, to explicitly acknowledge these ongoing debates and clarify that our model does not aim to resolve such molecular details (see the list of assumptions in the general response). The observed lifetime changes we attribute to qE reflect

PsbS- and pigment-dependent quenching, and are represented abstractly in our model via a schematic (Fig. 1A).

In addition, LHCII aggregation is known to give rise to far-red fluorescence emission that has been attributed to Chl-Chl charge-transfer states²⁷ and also is observed *in vivo* during qE (Miloslavina et al., Quenching in *Arabidopsis thaliana* Mutants Lacking Monomeric Antenna Proteins of Photosystem II*. *Journal of Biological Chemistry* **286**, 36830–36840 (2011); Marulanda Valencia, W. & Pandit, A. Photosystem II Subunit S (PsbS): A Nano Regulator of Plant Photosynthesis. *Journal of Molecular Biology* **436**, 168407 (2024).). As noted in the general response, our model does not explicitly consider Chl-Chl interactions, nor do we consider any specific physical mechanism of quenching. If Chl-Chl interactions or other mechanisms contribute to quenching of Chl excited states, they will contribute to the decrease of fluorescence lifetime, and therefore be captured by the TCSPC measurement. Their effects might be captured empirically by our framework, but not mechanistically resolved.

3. That camp tends to believe that zeaxanthin, and perhaps other carotenoids, in a yet unclear and perhaps more indirect way affect this change. The jury is still out on this subject but the authors do not even mention that their assumption is not accepted by many other experts in the field. Since the whole ms rely on this assumption, the fact that it is not consensus cannot simply be ignored. I myself do not know what to believe in – maybe the view of the authors is true – but a ms on the subject in a highly respected journal should not without comments/discussion rely on assumptions that many others think are wrong.

Response: As mentioned in the general response, our kinetic model neither assumes direct quenching by carotenoids nor excludes indirect pathways. To clarify, we now explicitly acknowledge the lack of consensus in the field and the range of mechanistic interpretations in our revised manuscript. We also include additional citations to relevant literature from differing viewpoints, as shown in response to Comment #2 of Reviewer #1.

4. In addition:

- Related to what written above: Since only a direct quenching to the electronic ground state is considered, and a “Stern-Volmer” analog description is used to describe it, doesn't the design of this study simply exclude any other possible quenching models?

Response: Please see assumption 3 in the general response. The Stern-Volmer-type relationship in our model simply describes a process (not necessarily returning to the ground state) that competes with the normal fluorescence decay and is proportional to the concentration of a quencher. It is intended as a phenomenological tool, describing the correlation between quencher concentration and fluorescence lifetime. It does not imply a specific molecular quenching mechanism. This approach was chosen for its simplicity, interpretability, and experimental

tractability. We clarify this in the revised manuscript and also cited foundational work⁴⁵ that support this description in modeling.

(Main text Page 8 Line 160-163) “While this phenomenological formulation can include multiple alternative quenching mechanisms, it enables effective kinetic fitting of fluorescence lifetimes, and it was chosen for its simplicity, interpretability, and capacity to capture the dynamic contributions of multiple quenching species.”

5. An interaction between the excited state of the quenched Chl-protein complex and the quencher molecule could give rise to a new fluorescing species with different emission spectrum and lifetime. Fluorescence is detected at only wavelength (680 nm) and the kinetic analysis only rely on average lifetimes, wouldn't such a species be invisible in this analysis?

Response: We thank the reviewer for raising this point. Our detection wavelength is centered at 680 nm to monitor Chl a fluorescence. The excited state species described by reviewer 3 would reduce excited Chl a population, as a competing pathway that is incorporated into Stern-Volmer expression. Thus, any mechanism that reduces Chl a fluorescence will be captured in our lifetime measurements. The use of amplitude-weighted average lifetime at 680 nm represents a collective effect resulting from quenching of Chl a by all the species and quenching pathways, we revised the Methods section to clarify this.

(Main text Page 22-23 Line 511-515) “Note that the measured chlorophyll fluorescence decay results from the averaging of tens of thousands of rate constants. As we showed in previous work⁶⁵, a decay curve arising from 30,000 rates can be perfectly fit by a sum of two exponentials, though these components do not correspond to any specific underlying rates or combinations thereof. The amplitude-weighted average lifetime values obtained from each snapshot were directly used in constructing and testing the NPQ model.”

6. Also concerning quenching mechanisms: In addition to the ones discussed and analysed here, qH was identified in the lab of the authors. It has also been discussed that spillover from PSII to PSI may occur (for example in <https://academic.oup.com/pcp/article/64/8/858/7148152>, <https://www.biorxiv.org/content/10.1101/2024.04.18.590023v2> and <https://www.biorxiv.org/content/10.1101/2025.01.26.634902v1>). Cyclic electron flow may also influence NPQ. Do the authors believe that these processes are irrelevant for their analysis? If so, motivate.

Response: We thank the reviewer for this thoughtful question. We do not claim that these processes are irrelevant to NPQ in general. However, based on our experimental design and some prior testing, their contribution was found to be negligible under the conditions used in this study. For example, we previously examined qT using *stn7* and *npq4stn7* mutants in Arabidopsis leaves and observed no significant differences in fluorescence lifetimes within our timescale (shown in Lam et al.¹⁸ and some unpublished data). qT tends to contribute more significantly to

algae (Steen et al., Interplay between LHCSR proteins and state transitions governs the NPQ response in *Chlamydomonas* during light fluctuations. *Plant, Cell & Environment* **45**, 2428–2445 (2022).).

qH is activated under cold and high-light stress, which does not apply to our growth or experimental conditions²¹⁻²². Our main purpose in this study is to provide a quantitative but simplified NPQ model to eventually help provide guidance on understanding the kinetic behaviors of the key quenching components as well as crop yield improvement, therefore making qE, qZ, and qI the key components to incorporate, to balance between predictability of the model and the conciseness and usability of the model.

In addition, both qT and CEF are unlikely to contribute considerably under the high light intensity and short timescales relevant to this study (Ramakers et al., Unravelling the different components of nonphotochemical quenching using a novel analytical pipeline. *New Phytologist* **245**, 625–636 (2025)). While cyclic electron flow can influence ΔpH , its impact would likely be similar across all mutants tested, as PSII was saturated by the laser pulses.

We now clarify this rationale in the Introduction of the revised manuscript.

(Main text Page 4 Line 63-70) “Since qT does not directly quench ¹Chl*, it contributes minimally to the PSII Chl *a* fluorescence quenching in the organism studied here, *Nicotiana benthamiana*, as supported by prior tests showing negligible qT-associated changes in ¹Chl* fluorescence lifetimes under our experimental conditions¹⁸ ... The most recently discovered chloroplast lipocalin-dependent sustained antenna quenching (qH) occurs under cold and high light conditions²⁰⁻²², and is therefore unlikely to be relevant under the typical growth and experimental conditions used for *N. benthamiana* or target crops.”

(Main text Page 6 Line 126-127) “we construct a lifetime-based quantitative kinetic model of NPQ focused on the largest components in land plants: qE, qZ, and qI.”

7. Is there evidence that all these postulated protein PX complexes are formed in the membrane, and that the action mechanism for at least some of the carotenoid is only indirect in the membrane, without forming such PX complexes?

Response: We appreciate this insightful question. In our model, PX and QX represent abstract states of pigment–protein association and activation, rather than asserting specific structural assemblies. Beyond the proportionality of NPQ to the concentration of xanthophylls, the PX → QX transition reflects the requirement for quenching activation via protonated PsbS. No further assumptions are made as to the molecular mechanism of xanthophyll’s role or the mechanism by which PsbS activates quenching.

In addition, while carotenoids can indeed form protein-bound complexes and act as direct or indirect quenchers^{6,24-26,30-35}, membrane-dispersed carotenoids primarily function as antioxidants and likely contribute minimally to NPQ on the timescales studied here (eg., Carotenoids. eds. Wise, R. R. & Hooper, J. K., (Springer Netherlands, Dordrecht, 2006). doi:10.1007/978-1-4020-

4061-0_16; Strzalka et al., Carotenoids and Environmental Stress in Plants: Significance of Carotenoid-Mediated Modulation of Membrane Physical Properties. *Russian Journal of Plant Physiology* **50**, 168–173 (2003).). In contrast, carotenoids located in close proximity to Chl within protein complexes are thought to play a more significant role in quenching (e.g., Liguori et al., From light-harvesting to photoprotection: structural basis of the dynamic switch of the major antenna complex of plants (LHCII). *Sci Rep* **5**, 15661 (2015)).

8. More than 45 parameters - quenching efficiencies for various carotenoids, interconversion rates etc. - are fitted in the model. Doesn't this mean a severe risk of overfitting given the amount of independent experimental data? And what does this mean for the error range of the parameters? The number are sometimes given in tables with five digits, which cannot be appropriate. These aspects should be discussed.

Response: We appreciate this concern. While the model contains multiple parameters, most are shared across genotypes, and the model is trained on diverse datasets across mutants and light sequences. Importantly, it was able to accurately predict behavior in genotypes that were not used for parameter fitting with the same set of parameters, demonstrating generalizability. We have added a sensitivity analysis (based on refitting the experimental data and computing bootstrapped 95% confidence intervals) to identify which parameters most influence model behavior (**Table S1** and **Table S2**). We reduced the number of significant digits in our parameter tables to align with these error margins. We have found that in general, the lower and upper bounds of the parameter set are within an acceptable range and increase our confidence in our mean fitted values.

9. A 1 mW beam of laser pulsed measuring light is used. I assume that the same leaf spot has been exposed for ca 20 minutes. If I understand it correctly, this measuring light is much higher than the actinic light which would explain why even the npq1/npq4 mutant has quite substantial quenching (ca 0.7). It is likely that in wt and other mutants, the measuring light may cause even bigger effects. I think the authors need to discuss this issue.

Response: Each snapshot involves 1 second of laser exposure, acquired every 15 seconds (83 total over 20 minutes). We revised the Methods section to clarify these technical details.

(Main text Page 22 Line 497-500) “Fluorescence decay profiles were recorded with a laser excitation and detector integration time of 1 second (termed a "snapshot"), collected every 15 seconds to ensure sufficient temporal resolution. A total of 83 snapshots were taken over any type of the 20-minute sequences, including 3 initial snapshots in the dark-acclimated state.”

(Main text Page 23 Line 514-516) “The amplitude-weighted average lifetime values obtained from each snapshot were directly used in constructing and testing the NPQ model. Each lifetime thus corresponds to a single data point in the figures.”

To quantify this effect, we performed control measurements using laser exposure without actinic light and modeled the result (Figure S3). Interestingly, the qI model with fitted parameters from actinic-light experiments accurately predicted the laser-only quenching baseline, suggesting that this slow component is accounted for. The contribution of the qI in various mutants was shown in Figure 4 and Movie S1.

10. The model performs well for WT under fluctuating light but less well under the 3HL-1D-1HL-3D-9HL-3D regime, or for example in *zep2*. What is the explanation for this?

Response: Given the number of experiments needed to collect these data, it was necessary to use different batches of plants. We consider that this discrepancy likely reflects biological variability across experimental batches, as we noted in the manuscript. The 3HL-1D-1HL-3D-9HL-3D sequence was intentionally designed to be more complex, and subtle differences in pigment pools or enzyme levels may contribute to the observed variation. We have already elaborated on this point in the Discussion in the original manuscript.

(Main text Page 18 Line 401-405) “Biological variation introduces uncertainty in parameter estimation, as differences in growth conditions and leaf age affect pigment pools, enzyme activity, and consequently, fluorescence lifetime and xanthophyll interconversion rates—even within the same genotype. These factors likely contribute to the model’s slight NPQ underestimation in the 3HL-1D-1HL-3D-9HL-3D sequence (**Fig. 3C**).”

11. The model assumes a halved number of pigment binding sites in the *lut2* mutant to approximate LHC structural changes. However, alterations in VAZ binding affinities and LHCII trimer stability may occur in the absence of lutein may have nonlinear effects on NPQ kinetics.

Response: We agree that the removal of lutein may cause multiple structural and kinetic changes. In the model, [P]_{tot} in *lut2* was fitted separately from [P]_{tot} in WT. Additionally, the rate constants associated with pigment binding and dissociation were separately fitted in *lut2* and were found to differ from WT; in particular, the X \rightleftharpoons PX equilibrium shifts to the left for all xanthophylls, consistent with reduced LHCII trimer stability, and the ratio of binding affinities between V, A, and Z differs in *lut2* versus WT, consistent with alterations in VAZ binding affinities. We have clarified this point in the main text.

(Main text Page 11 Line 250-254) “The rate constants associated with pigment binding and dissociation were found to differ in *lut2*; in particular, the X \rightleftharpoons PX equilibrium shifts to the left for Vio, Ant, and Zea, consistent with reduced LHCII trimer stability, and the ratio of binding affinities between Vio, Ant, and Zea changes from approximately 2:3.5:7 to 1:1:3, consistent with alterations in VAZ binding affinities.”

While more detailed modeling of structural compensation is possible, we chose a minimal approach for tractability given the main goal of our study.

12. The model also assumes that quenching contributions are additive across mechanisms which may not necessarily be true. I guess this could be tested directly by comparing the modelled sum of two single mutant behaviours with the observed double mutant behaviour. Discrepancies here could reveal pleiotropic interactions or structural compensation that the model does not account for. Without this validation, the assumption of linear additivity remains a speculation.

Response: As noted in the general response, our model does not assume additivity a priori. Rather, it was a conclusion drawn from the model's ability to predict wild-type and double mutant behavior from integrating each of the relevant quenching components, with parameters obtained from the five essential genotypes used for fitting.

13. The authors talk about a “lifetime-based quantitative kinetic model that disentangles individual photoprotection components” and lifetime snapshots have been taken. But as far as I understand in their model they only use average lifetimes, not the multi-exponential kinetics. Is it then really a lifetime-based quantitative kinetic model?

Response: We quote Papageorgiou and Govindjee (Barbara Demmig-Adams et al., *Non-Photochemical Quenching and Energy Dissipation in Plants, Algae and Cyanobacteria*. (Springer Netherlands, Dordrecht, 2014), p21.) “Note that change in fluorescence intensity can be simply due to change in the concentration of Chl; thus, measurement of fluorescence lifetime is crucial in reaching firm conclusions.”, the Chl fluorescence decay results from averaging together many tens of thousands of rates. As we showed in previous⁵⁷, a decay curve arising from 30,000 rates can be perfectly fit by two exponential components but the two decay components have no relation to any underlying rates of combination of rates. The weighted average lifetime is directly proportional to the fluorescence yield and seems the most reasonable and feasible way to represent the overall degree of quenching. We revised our manuscript accordingly, please refer to our responses to Comment #5 of Reviewer #3.

14. As lifetimes are proportional to the fluorescence yield, wouldn't it be possible to reach the same or similar conclusions from steady-state fluorescence measurements, just that the calculated rates in that case would be proportional and not absolute? If so, this analysis does really not make use of the extra information obtained from ultrafast time-resolved experiments.

Response: Our model is indeed based on fluorescence lifetimes, which are directly proportional to the absolute fluorescence quantum yield. As such, both snapshot lifetime measurements and hypothetical snapshot quantum yield measurements would, in principle, be equally valid for the modeling framework we present. However, at present, there is no technique available that allows true snapshot quantum yield measurements with sufficient time resolution.

We believe the reviewer may be referring to steady-state fluorescence yield measurements obtained via PAM fluorometry when suggesting that similar results could be obtained without

lifetime data. While PAM is widely used and powerful for many applications, it differs from our approach in key ways:

1. PAM measures relative fluorescence changes (e.g., with F_0 , F_v/F_m) rather than absolute fluorescence quantum yield value, and therefore these fluorescence changes could not be proportionally convertible with lifetime.
2. PAM is sensitive to the concentration of Chl which could be affected by many factors across measurements on various samples. Lifetime does not.
3. PAM NPQ parameters do not scale linearly with absolute quenching capacity at high NPQ levels, limiting their interpretability in the quantitative kinetic model.

15. The same lab has previously modelled the photoprotection in *Nannochloropsis oceanica*. In the introduction of this ms they lists a number of “significant limitations” of that model due to various assumptions. Although I acknowledge the skills of the lab and the complexity of this analysis, I am worried that some of the assumptions they have made here, and the issues mentioned above, reduces the value of this ms and the validity of the claims. Maybe the authors can present good explanations and good discussion on these points, but I am quite sceptic.

There are many papers on NPQ mechanisms which suffer from the same weakness; making assumptions that others do not buy. It is not easy when there is no consensus in the field about general assumptions but to move forward it is important to interpret data in the light of the whole body of literature, and this is where I think this ms fall short. Therefore I cannot recommend publication, at least not in the current form.

Response: We appreciate the reviewer’s candid feedback and acknowledge the importance of critically evaluating modeling assumptions, especially in a field where mechanistic consensus remains elusive. We hope the lists of assumptions made and not made along with the substantial clarifications to the manuscript adequately address these concerns.

Reviewer #4 (Remarks to the Author):

In this submission, the authors attempt to use a fluorescence lifetime modelling approach to understand the contributions of different components of NPQ in wildtype and mutant *Nicotiana benthamiana* plants. This review is written from the perspective of someone who has spent a lot of time measuring NPQ in whole plants but using PAM-style approaches and with little experience of modelling. I am therefore reading this as an interested reader rather than an expert and my review is delivered on this basis. I assume more expert reviewers have also been consulted.

Response: We thank the reviewer for their thoughtful and constructive feedback, and especially for engaging with our manuscript from the perspective of an experienced NPQ researcher with a

PAM-based background. We appreciate the suggestions that have helped us clarify and improve the manuscript. Below we address each of the reviewer's comments in turn.

1. The paper opens by mentioning the existence of 5 mechanisms of NPQ, though I was a little unclear from the introduction what they consider these to be. I assume that they are identifying Lut quenching as a distinct NPQ mechanism, or are they including this in qE, not quite clear from the wording.

Response: We appreciate the reviewer highlighting this point. In the revised manuscript, we have clarified that the five types of NPQ referenced are: energy-dependent quenching (qE), zeaxanthin-dependent quenching (qZ), photoinhibition (qI), state transitions (qT), and sustained quenching (qH). Among these, our model includes only qE (quenching by both VAZ and lutein), qZ, and qI, as they account for the dominant NPQ responses under the conditions tested. In other words, lutein-dependent quenching is treated as a subcomponent of qE. We have revised the Introduction to more clearly distinguish between the types of NPQ included in our model and those excluded, along with brief justifications. As noted in the general response, we also revised our use of the term "mechanism" to "types of NPQ" or "NPQ components" (e.g., qE, qZ, qI) to avoid misunderstanding.

(Main text Page3-4 Line 49-70) "To date, five distinct types of NPQ have been discovered, with various genetic mutants deepening our understanding of land plant NPQ at the molecular level. Among these types of NPQ, energy-dependent quenching (qE)... The longer-lasting Zea-dependent quenching (qZ) functions independently of PsbS... state transitions in light-harvesting complex II (qT)... Since qT does not directly quench ¹Chl*, it contributes minimally to the PSII Chl *a* fluorescence quenching in the organism studied here, *Nicotiana benthamiana*, as supported by prior tests showing negligible qT-associated changes in ¹Chl* fluorescence lifetimes under our experimental conditions¹⁸. The most slowly relaxing NPQ component, photoinhibition (qI)... The most recently discovered chloroplast lipocalin-dependent sustained antenna quenching (qH) occurs under cold and high light conditions²⁰⁻²², and is therefore unlikely to be relevant under the typical growth and experimental conditions used for *N. benthamiana* or target crops."

2. Also, I am not clear which are the canonical and non-canonical mechanisms.

Response: We thank the reviewer for raising this point. In our initial wording, we referred to qE and qZ as "canonical NPQ mechanisms" because they are well-defined in terms of their molecular actors, specifically carotenoids such as zeaxanthin and lutein, have well-established genetic and biochemical bases, and account for the vast majority of detectable NPQ under the short actinic light regimes used in this study. However, we now realize that this terminology may be confusing. To avoid ambiguity, we have revised the manuscript to clarify our use of terminology.

(Main text Page 3-4 Line 61-63) “Unlike qE and qZ, which involve specific carotenoids, state transitions in light-harvesting complex II (qT) regulate light harvesting by redistributing excitation energy between photosystem I (PSI) and photosystem II (PSII)¹⁷.”

3. And is reaction center quenching, associated with Ca²⁺ release, included in your count or canon?

Response: We thank the reviewer for raising this aspect. While our model focuses on qE, qZ, and qI, we do not exclude the possibility that other mechanisms—such as reaction center quenching—may contribute under certain conditions. If such a mechanism leads to detectable quenching of Chl *a*, it would be reflected in the observed lifetime decay and grouped under the broadly defined photoinhibition category qI, as it occurs on a similar timescale. However, as noted in the general response, our current model does not aim to mechanistically resolve these processes, and therefore does not explicitly include Ca²⁺-dependent RC quenching. We have added clarification in the revised manuscript explaining that the model intentionally simplifies the system to enable quantification of dominant, well-characterized NPQ components. Please refer to Response 6 to Reviewer 3.

4. Anyway, they then go on to develop a model that only considers 3 mechanisms of quenching. This could perhaps be more clearly explained in the introduction.

Response: We agree with this comment and have tried to be more explicit in the Introduction. The primary aim of our study is to develop a quantitative yet simplified model of NPQ that captures the essential kinetic features of major quenching components, with the broader goal of informing strategies for improving crop productivity. To achieve a balance between biological accuracy and model usability, we focused on the three dominant NPQ components—qE, qZ, and qI—which account for the majority of dynamic quenching responses under our experimental conditions. Other mechanisms, such as qT and qH, were excluded based on their minimal contributions in our system and timescale, as well as to maintain conceptual clarity and tractability. We have revised the Introduction to make this scope and rationale more explicit, as also detailed in the previous comments.

5. The main experimental approach involves the measurements of the decay kinetics of fluorescence under different conditions. I was a little unclear about the details of how these measurements were made, but I assume this is measuring the fluorescence decay following application of 404 nm flashes (duration?) and these decays were fitted with a bi-exponential decay function.

Response: We appreciate the opportunity to clarify this aspect of our methodology. Full details of the fluorescence lifetime measurements are now more clearly presented in the revised Methods section titled “*Whole-leaf Fluorescence Lifetime Measurements and Analysis.*” We have revised this section to enhance clarity and ensure that the experimental setup is more accessible to readers unfamiliar with time-resolved techniques. Please also refer to our responses to Comment #9 of Reviewer #3.

6. It is not clear how good the fit is across different plants and conditions, as no examples are shown.

Response: The fitting of fluorescence decay curves is a well-established method in our laboratory⁶⁴, where we compared snapshot-based and PAM-based measurements). The quality of the fits is maintained across all the genotypes by collecting a consistent number of photons per snapshot, ensuring robust signal-to-noise ratios. Additionally, the short instrument response function (~36 ps) of our setup provides sufficient time resolution to resolve fluorescence lifetimes ranging from several nanoseconds to approximately 100 picoseconds across all genotypes under all the conditions examined.

In response to the reviewer’s suggestion, we now include a representative example of fluorescence decay fitting in the Supplementary Information (**Fig. S7**) to illustrate the quality of the fits.

(Main text Page 23 Line 516-517) “A representative example of the raw fluorescence decay profile, bi-exponential fit, and IRF is shown in Figure S7.”

7. The overall strategy of the modelling made sense to me – mutants with varying lesions in NPQ were fitted using their model and the parameters estimated were used predictively to compare to wild type and further mutants including in additional light regimes. It is always a concern with modelling that it is always possible to find a model to fit data, but using this in a predictive way gives confidence. The initial fits and predictions are not perfect but seem to capture the main processes occurring. Given the number of simplifications required this is reasonably convincing.

Response: Indeed, one of our main goals was to test whether a simplified kinetic model, grounded in Chl fluorescence lifetime and pigment dynamics, could capture the essential features of NPQ behavior across genotypes and conditions. We are encouraged that the overall strategy was clear and convincing to the reviewer.

8. In terms of applying the models to predict the optimal solution for optimising NPQ, I was not totally sold on the usefulness of the approach, though I think this is a useful thought experiment.

Response: We appreciate the reviewer’s thoughtful and cautious perspective. The simulations involving overexpression of VDE, PsbS, and ZEP were intended as a conceptual

demonstration—an initial attempt to illustrate the predictive potential of our model. We also fully acknowledge that the current version of the model has limitations, particularly in its applicability to complex and variable natural conditions. Our goal in including these simulations was not to make definitive physiological claims, but rather to explore how the model might be used to guide hypotheses and inform experimental design. As acknowledged in the Discussion, experimental validation of these predictions remains an important next step. We are actively working on this in ongoing and future studies. Please refer to Response 6 to Reviewer 1.

9. In the discussion, the authors are honest about the limitations of their approach, especially in terms of the need to consider biological variability. Whether this experimental approach will ever be as accessible as traditional PAM style measurements I do not know, but this does have the potential to provide useful information in at least some circumstances.

Response: We fully acknowledge the practical limitations of our current model and experimental approach, and we remain committed to further refining the methodology to improve its usability and relevance under more complex and natural conditions. While our model is fundamentally based on chlorophyll fluorescence lifetime measurements, which offer distinct advantages in quantifying NPQ, we also fully recognize that PAM-based approaches are more widely adopted in the field. Bridging the gap between these two types of measurements is an important future goal, and we hope that continued development of our model methodology might facilitate greater interoperability and accessibility for broader applications.

Reviewer #5 (Remarks to the Author):

In this study, Lam et al. explore the specific contributions of NPQ components in *Nicotiana benthamiana* under fluctuating light conditions. By integrating chlorophyll fluorescence lifetime measurements on intact leaves with HPLC pigment profiling, the authors construct a kinetic model that quantifies the activity and molecular efficiency of distinct mechanisms (qE , qZ , qI) and of individual xanthophylls (Zea, Lut, Ant). Through the use of single and double mutants affecting specific NPQ pathways (some pre-existing and others generated in this study), they deconvolute overlapping processes and fit the model through parameter optimization based on experimental fluorescence decay data. The model computes the contributions of each quencher to energy dissipation over time and accurately predicts NPQ dynamics under complex light regimes. This represents an important methodological advancement, offering a predictive framework to guide genetic strategies aimed at improving light use efficiency and photoprotection.

Response: We thank the reviewer for recognizing the methodological novelty and predictive potential of our kinetic model. We also appreciate the constructive suggestions and fully agree that the biological complexity of NPQ requires further refinement of our framework. Below, we respond in detail to each of the reviewer's comments.

1. Nevertheless, many biological complexities are simplified. The fluctuating light conditions were selected a priori, and the parameter values used in the model may be influenced by various factors.

Response: We agree that our current model simplifies certain aspects of natural light environments. To partially address this, we tested the model across a range of light sequences, including both continuous (20HL), regular (5HL-10D-5HL) and irregular (3HL-1D-1HL-3D-9HL-3D) regimes, designed to capture variation in NPQ activation and relaxation kinetics contributed by different components. These were selected to provide controlled but progressively more challenging conditions for model prediction. We acknowledge that in natural environments where light conditions are highly unpredictable, further refinement is needed, and we added a few more sentences to discuss this limitation explicitly in the revised manuscript (Discussion section).

(Main text Page 18 Line 407-418) “In addition, the model also simplifies the illumination treatments by using three designed regular and irregular HL-to-D illumination sequences. These were chosen to capture variation in NPQ activation and relaxation kinetics contributed by different components, with progressively more challenging conditions for model prediction, providing a controlled platform to interrogate NPQ mechanisms before extending the model to more complex light regimes. However, natural sunlight exhibits more unpredictable and unpatterned HL-to-low-light (LL) fluctuations⁵⁸, and more gradual transitions ... Incorporating more stochastic illumination patterns, HL-LL transitions, and light intensity-dependent rate constants may improve the physiological relevance of this framework for understanding NPQ regulation under natural light fluctuations, if not offset by increased phenotypic variability.”

2. For example, in natural environments characterized by unpredictable sunflecks and alternating low/high light conditions, the model could potentially underestimate or overestimate NPQ dynamics. Or even the approximations used for qI. In fact, the authors themselves acknowledge the main limitations (lines 362 to 376). Nonetheless, the model is coherent with experimental data and provides a solid foundation for further refinement.

Response: We agree with the reviewer that this is a significant consideration. Indeed, environmental light variability introduces layers of complexity that our current model does not yet fully capture. We are fully aware of this limitation as we described in our discussion. Incorporating light intensity dependence would increase physiological realism but also significantly increase model complexity and experimental burden. Nevertheless, incorporating field-realistic light inputs and further parameter refinement are important next steps for advancing the model’s applicability, and remain a priority for our future work.

3. In the natural environment, the sudden changes in light intensity are accompanied by strong

variations in the spectra of actinic light. A short paragraph could be dedicated to this factor of complexity and its effect on measurements.

Response: In our understanding of responses to spectral changes (change of UV A/B and far-red light amplitude) accompanying sun flecks, the response of plants such as shade avoidance occurs on longer timescales than those studied in our work. (Sellaro et al., Making the most of canopy light: shade avoidance under a fluctuating spectrum and irradiance. *J Exp Bot* **76**, 712–729 (2024). gives a recent review)

4. Additional points:

- A sensitivity analysis of kinetic parameters (k) across NPQ components would improve robustness and highlight critical points to be implemented to cope with large biological variability.

Response: We appreciate this recommendation. In response, we have performed a sensitivity analysis of all kinetic parameters and included the results (via refitting the data several times and bootstrapping 95% confidence intervals) in the revised Supplementary Information (Table S1 and S2). This analysis identifies which parameters most strongly influence the model's outputs (indicated by a narrow range between lower and upper bound) and highlights potential targets for future refinement or experimental focus. We are grateful for the reviewer's insight, which we believe has strengthened our study.

5. In Fig. S3, quenching induced by the laser in the absence of actinic light reaches ~0.6 ns after 20 minutes, which is non-negligible. How frequently are lifetime snapshots taken per induction curve? Reducing snapshot density might minimize laser pulse-induced quenching, especially during dark phases, and improve the accuracy of recovery kinetics; a potential trade-off.

Response: The laser exposure and signal integration time is 1 second (termed a "snapshot"), represented by each of the data points in the figures. The frequency of the measurement is 1 snapshot per 15 seconds. We revised the Methods section to clarify more technical details of the measurements. Please refer to our responses to Comment #9 of Reviewer #3.

We agree that laser-induced quenching is non-negligible and have performed laser-only controls, which confirmed this. We agree that even reducing snapshot/measurement frequency could help reduce this laser-induced quenching, and we did various testing that proves this. However, even if we reduce the frequency, the laser baseline reduces but still is not negligible. We therefore decided to use "1 snapshot per 15 s" measurement frequency to prioritize NPQ dynamic resolution, and incorporate laser baseline, which turned out to be qI, in our model to account for this phenomenon.

6. Line 334: please verify consistency with respect to Fig. 4A. The major components are Lut and Ant, not Vio.

Response: We thank the reviewer for catching this inconsistency. We have revised the corresponding description for Fig. 4A in the main text to accurately reflect that Lut and Ant are the dominant contributors between t=0 and t=3 min, not Vio.

(Main text Page 16 Line 369-371) “In WT, from t=0 to t=3 min (**Figure 4A-B**), Lut and Ant are the major quenching contributors due to their rapid activation capabilities. Between t=3 and t=5 mins, quenching contributions from Ant and Zea are similar in magnitude.”

RESPONSE TO REVIEWER COMMENTS

For clarity, we have color-coded this response letter as follows:

Black: Reviewer comments

Blue: Our responses

Red: Direct citations of the manuscript (Line numbers refer to the no tracked-changes version; page numbers are consistent across both versions.)

Reviewer #1 (Remarks to the Author):

The authors replied to my comment point-by-point:

1. Choice of mutants

The authors argue that thylakoids are more suitable for transient absorption (TA) measurements, and I am prepared to accept that point in principle. However, the material has not been properly characterized with respect to electron transport and the resulting proton gradient (ΔpH) that drives NPQ. Both the kinetics and amplitude of these processes should be reported. Without such data, neither the measurements nor the resulting model can be considered reliable. Furthermore, reference 26 does not, in my view, provide higher-quality data than comparable work in Arabidopsis, nor does it present convincing evidence that zeaxanthin directly receives excitation energy from chlorophyll.

Response: It is well established that ΔpH generated through electron transport is a primary driver of NPQ induction. This relationship has been validated in numerous studies across model systems, including both Arabidopsis and Tobacco species. For example, Cardol et al. (2010, *Biochimica et Biophysica Acta*, doi:10.1016/j.bbabi.2009.10.002) explored the influence of proton motive force across the thylakoid membrane on NPQ onset in tobacco; Miyake et al. (2005a, *Plant Cell Physiology*, doi:10.1093/pcp/pci067) examined the role of PSI/PSII electron fluxes in determining NPQ magnitude; and Miyake et al. (2005b, *Plant Cell Physiology*, doi:10.1093/pcp/pci197) investigated the effects of light intensity on cyclic electron flow and its relationship to NPQ. Therefore, our work builds upon this foundational understanding, rather than re-demonstrating it specifically for *N. benthamiana*.

Also note that the kinetics of ΔpH activation are considered fast enough on the timescale of our experiment to be effectively instantaneous, hence ΔpH is phenomenologically modeled as an on/off switch of constant amplitude. Since all experiments are performed at the same light intensity, this is a reasonable simplifying assumption to make to avoid overparameterization of the model.

We would also like to re-emphasize that our model does not presuppose a specific physical mechanism of Zea-mediated quenching. Instead, it captures the kinetics of NPQ components in a manner consistent with the availability of individual quenchers. We explicitly state that Zea's role in our model should be interpreted functionally, based on the timing and extent of its

contribution to quenching, rather than as evidence of a particular photophysical mechanism such as direct energy transfer from Chl to Zea.

With respect to the spectroscopic evidence, we note that high-quality data demonstrating direct Zea-mediated quenching via excitation energy transfer through the Qy band are limited even in Arabidopsis. If the reviewer does not find reference 26 sufficiently convincing, we respectfully submit that, to our knowledge, no higher-quality spectroscopic dataset currently exists in Arabidopsis or in equivalent/more intact systems.

2. Zeaxanthin as a quencher

The authors appear unwilling to engage with the full scope of research on the relationship between zeaxanthin and NPQ published over the past few decades. Instead, they continue to promote the long-standing and now outdated view originating with Demmig-Adams et al. some 35 years ago. This selective treatment of the literature risks perpetuating a simplified narrative that does not reflect current understanding.

Response: We thank the reviewer for pointing out the alternative theories of zeaxanthin's contribution to quenching. In response, we have added the following sentences to the Introduction of the main text:

(Main text Page 5 Line 81-85) “Other models propose, however, that carotenoids function not as direct quenchers, but as allosteric effectors that induce conformational changes in LHCs to facilitate energy dissipation³⁶⁻³⁸. Structural and spectroscopic studies support Zea-dependent but structurally mediated pathways contributing to quenching, suggesting that Zea may enable or stabilize specific protein conformations that facilitate NPQ without directly quenching Chl excited states^{36,39}.”

We would also like to clarify (as discussed in the first response letter and in *Model Assumptions* in the Extended Methods) that our model assumes the quenching rate constant depends linearly on the concentration of QX for qE or the concentration of Zea for qZ. This framework is agnostic to the specific mechanism by which carotenoids contribute to quenching, whether through direct energy transfer or through indirect mechanisms such as allosteric modulation. To make this clearer for readers, we have revised our wording throughout the entire manuscript. For example, we changed “quenchers” to “contributors to quenching” or “effectors of quenching” and “quenching efficiency” to “quenching effectiveness,” to better reflect the agnostic nature of our model.

Additionally, in response to this comment and to Comment 5, we apologize for omitting prior work on modeling NPQ. We found a number of relevant references (43–50; see the revised manuscript's Reference list) which are now included in a new paragraph in the Introduction as follows:

(Main text Page 5 Line 96-100) “There have been a number of previous modeling studies of NPQ kinetics, and qE in particular⁴³⁻⁵⁰. These models have generally been based on the VAZ cycle, with Zea and protonated PsbS acting as the sole quenching-

related components, most commonly represented by the 4-state 2-site quenching model proposed by Jahns and Holzwarth¹⁴, which takes into account both open and closed RCs and quenched and unquenched PSII antennas.”

We discuss the connections between the current work and references 43-50 in a new paragraph in the Discussion Section:

(Main text Page 15 Line 326-341) “In this work, our experimental protocol closes the RCs. We allow each component of the VAZ cycle to act as an inducible contributor to quenching, with differing activation rates and quenching effectiveness. We also include qI and qZ as well as the contributions of Lut, which is an important contributor to qE at short timescales and the major component for the *npq1* mutant which was also studied by Morales et al.⁴⁵. As Stirbet et al. point out⁵⁰, several existing models assume that the VAZ cycle (in particular, Zea) and PsbS protonation are the only components involved in qE quenching (the “4 state 2-site” model), simplifying NPQ dynamics but reducing the ability to capture complex photoprotective systems. These models also commonly make the simplifying assumption that the total pool of [Vio] + [Zea] is constant over short timescales. However, this limits the ability to compare kinetics across genotypes, which can exhibit large variations in the size of the total xanthophyll pool even after dark adaptation (**Figure S1, S2**). The memory effect explored by Matuszynska et al. is an intrinsic part of our model⁴³, as it was in our earlier algal model^{51,52}, and is unraveled in greater mechanistic detail here, demonstrating the importance of the Ant intermediate to NPQ memory during dark periods. Thus, our work here expands upon the existing body of qE modeling studies by providing a detailed molecular picture of photoprotection, connecting lifetime dynamics directly to specific molecular contributors and kinetic rates, and capturing multiple NPQ components (qE, qI, qZ).”

We provide further comments comparing existing models with ours in our response to Comment 5.

3. qI component

The model presented does not question the mechanistic origins of qI but rather addresses only its kinetic behavior. This raises the question: what is the real utility of such a model for physiologists? Without a mechanistic framework, a purely kinetic description of such a complex component offers limited biological insight.

Response: Indeed, the mechanistic basis of qI involves a complex and not yet fully understood set of molecular processes, including photodamage, D1 turnover, and possibly sustained pigment-protein aggregation. However, elucidating these mechanistic details lies outside the scope of our current study, which focuses on quantifying and separating the kinetics of overlapping NPQ components across genotypes.

Despite this, we argue that a robust kinetic characterization of qI offers substantial utility. Prior to this work, it has been challenging to disentangle qI contributions across different mutants or species, particularly due to its overlap with qE and qZ. Our model provides a framework to quantitatively resolve qI dynamics, as demonstrated in Figure 4 and Movie S1, which reveal distinct time-dependent contributions of qI in wild-type and mutants.

Importantly, isolating the kinetics of qI is foundational to the structure of our model. Without accounting for qI, it would not be possible to reliably separate the faster components, such as qE and qZ, especially since the qI contribution and onset vary significantly across genotypes and evolve over time. Assuming identical qI behavior in all mutants would introduce substantial inaccuracies. Thus, including qI in the model is essential to unlock more mechanistic insights downstream, such as the per-molecule quenching effectiveness of each component, the time-resolved NPQ contributions, and the predictive capacity for NPQ phenotypes across mutants, which constitute the central findings of our study.

In addition, a kinetic approximation of qI is also essential because the principle behind NPQ is that an increase in quenching will reduce photodamage (qI may not represent the entirety of photodamage, but remains a useful proxy). Thus, mutants with higher NPQ capacities will have lower accumulations of qI, as we see demonstrated in our model (Fig 4). Modeling the quantity of qI is thus not only useful for reliably capturing different NPQ kinetics but also provides an explanation for why mutants differ in NPQ recovery—some sustain more photodamage than others—and allows direct observation of the impact on NPQ on photoinhibition levels. We have added the following sentences to the manuscript to provide additional clarification.

(Main text Page 18 Line 405-406) “Thus, although qI is not modeled mechanistically, we are still able to observe the effects of NPQ activity on the accumulation of photodamage.”

(Main text Page 18 Line 414-416) “qI accumulates at an increased rate and acts as the major quenching contributor, explaining the lack of NPQ recovery in darkness”

Finally, we note that although the exact molecular mechanism of qI remains under investigation, our model provides a practical and extensible starting point for quantitatively analyzing qI using accessible chlorophyll fluorescence lifetime data. As new mechanistic details emerge, this modeling framework can be readily adapted to incorporate such refinements and may be helpful for testing various mechanistic hypotheses.

Regarding biological insight, we note that our earlier modeling work (Zaks et al., *PNAS* 2012) has been cited 183 times according to Google Scholar, and has been incorporated by others into physiological crop growth models such as BioCro. BioCro is a model that predicts plant growth over time given crop-specific parameters and environmental data as input (BioCro.org). We believe our current model will be even more useful, as it incorporates a more complete set of NPQ components, provides specific molecular contributions and biochemical rates from fluorescence lifetime-based modeling, predicts experimental results more accurately, and offers guidance for the design of overexpression mutants.

4. Kinetic aspects of NPQ

Even within the narrow kinetic focus, the treatment of NPQ dynamics in this manuscript remains incomplete. The relevant kinetic aspects of NPQ have not been adequately reviewed, and the analysis feels partial at best.

Response: See response to Comment 1 and 3. Without further specificity, it is difficult to respond to this comment. There are indeed entrenched positions on the mechanisms of NPQ, but without putting detailed models out for discussion by the broader community it is difficult to see a route to progress.

5. Suitability for Nature Communications

From the discussion surrounding the modelling work of Snellenburg et al., it is evident that this project is highly specialized and riddled with unresolved controversies. As such, it seems more appropriate for a specialist journal in photosynthesis research, rather than a broad-audience outlet such as Nature Communications. The current scope and accessibility do not justify publication in a general high-impact venue.

Response: Please see our response to Comment 2.

Turning specifically to the Snellenburg et al. work mentioned by Reviewer 1, the current paper and Snellenburg et al. are not contradictory, but rather they approach the task of modeling NPQ from entirely different angles. The emphasis of the Snellenburg et al. model is on characterizing the state of PSII using purely kinetic/mathematical parameters. They do not attempt to characterize biochemical pathways in any detail as that is not the goal of their study.

Snellenburg et al. modeled the evolution of PAM fluorescence by fitting population fractions and quantum yields to four PSII states. Their framework describes how much of PSII becomes quenched under given conditions. In contrast, our model reveals the involvement of different xanthophylls in promoting quenching and their relative effectiveness at the molecular level. By directly connecting lifetime changes to specific quenchers and kinetic rates, our approach provides a mechanistic picture of how photoprotection proceeds in photosynthetic systems.

By exciting Chl a with a saturating laser pulse we reduce the problem to one in which reaction centers are closed. If we didn't do that, the reaction centers would be either open or closed (an additional factor to model), in which case the work by Snellenburg et al. would become useful. Our assumption of reaction centers being closed is not an "unsolved controversy" but rather a direct result of our experimental design.

The work by Snellenburg et al. also makes assumptions, namely that the pool of xanthophylls is constant and importantly that the amount of violaxanthin and zeaxanthin are constant over the course of the experiment. This is a simplifying assumption, as the authors write: "the fitting results show that given the short duration of the experiments this extra complexity is not needed to adequately describe the data." The authors readily acknowledge that in reality, V to Z interconversion is taking place.

Thus, there is no contradiction or unresolved controversy between the two works, as Reviewer 1 seems to suggest.

While this level of molecular detail may seem unfamiliar to readers outside the photosynthesis community, the lifetime-based modeling framework itself is broadly applicable: it offers an intuitive and quantitative tool for disentangling dynamic quenching processes, even for a complex light-harvesting system. Moreover, by organizing these mechanisms into modular components, our manuscript helps general audiences understand how the photosynthetic apparatus functions as an integrated, dynamically regulated system.

Regarding the manuscript's suitability for *Nature Communications*, please also see our response to Comment 3 of Reviewer 3.

6. Authors' own admissions

The authors themselves acknowledge that their approach requires further experimental validation and, in all likelihood, improved quality in the kinetic fitting of their data. This further reinforces my view that the manuscript, in its current form, is not ready for publication in its intended journal.

Response: Please see response to Comment 5.

Reviewer #1 (Remarks on code availability):

n/a

Reviewer #2 (Remarks to the Author):

Photoprotection mechanisms collectively referred to as non-photochemical quenching (NPQ) remain incompletely understood due to the large number of underlying components and their diversity among species. In this study, the authors established a convenient technique to monitor chlorophyll fluorescence lifetime changes accompanying the induction and relaxation of NPQ under fluctuating light conditions. Furthermore, they proposed a simplified NPQ model, composed of only the essential elements, that accurately reproduces these fluorescence lifetime dynamics. I recommend the prompt publication of this manuscript.

Compared with the commonly used PAM-based parameter, $(F_m - F_m')/F_m'$, the newly proposed $\tau F(t)$ has two distinct advantages: it is less influenced by residual NPQ present before the onset of measurements, and it exhibits higher linearity with respect to changes in quencher concentration. The authors applied model fitting to fluorescence lifetime data obtained from both wild-type plants and multiple NPQ mutants. Beginning with a double mutant containing the fewest NPQ components and progressively incorporating additional factors, they ultimately determined the parameter set representing all NPQ factors in wild-type plants. The validity of this NPQ model was confirmed by its ability to predict, with high accuracy, the fluorescence lifetime dynamics of independent NPQ mutants under fluctuating light conditions. The authors further suggest that this model could provide insights into improving photosynthetic responses and potentially contribute to crop yield enhancement.

In constructing the model, only essential components were deliberately included, ensuring neither redundancy nor omission. Consequently, the model does not aim to fully replicate the complex behavior of NPQ in natural environments, nor does it present novel discoveries regarding NPQ mechanisms. The authors explicitly state that these were not the primary objectives of their work.

Since the inception of NPQ research over half a century ago, non-invasive PAM measurements have greatly advanced the field. The approach proposed in this manuscript—combining fluorescence lifetime measurements with a streamlined NPQ model—offers the potential to detect subtle alterations in minor NPQ components that PAM-based analyses cannot resolve. This advance is expected to be valuable for future photosynthesis research and for practical applications such as crop breeding. The publication and subsequent broad validation of this method across diverse environments and species may eventually establish it as an alternative to conventional PAM-based techniques.

Response: We thank the reviewer for their positive assessment of our work.

As the reviewer noted and we discussed in Sylak-Glassman et al. *Photosynth Res* 127, 69-76 (2015)⁷¹, PAM measures the total amount of fluorescence and may not always be related to the rate of fluorescence quenching, because of chloroplast movement, damage etc. In contrast, fluorescence lifetime measurements directly reflect the rate of excited-state quenching and can thus be quantitatively modeled using a kinetic framework like the one we present here. This same point has also been made by Papageorgiou and Govindjee, as quoted in our previous response (Barbara Demmig-Adams et al., *Non-Photochemical Quenching and Energy Dissipation in Plants, Algae and Cyanobacteria*. (Springer Netherlands, Dordrecht, 2014), p21.). We agree, however, that PAM measurements have been and will continue to be highly valuable for NPQ studies. We hope that continued development of our modeling methodology will facilitate greater interoperability and accessibility, helping to bridge the gap between these two types of measurements for broader applications.

Reviewer #3 (Remarks to the Author):

1. The ms has been modified in several ways to satisfy the concerns of me and the other reviewers. I think that the authors have done a good job in improving the explanation of what they have done and why. In particular, I appreciate the list of assumptions made and not made in the Extended methods section. But as not everyone immediately reads the SI, there should a specific reference to this information in the main text, there may be others that like me need this description.

Response: We appreciate the suggestion and have added a specific comment in the main text pointing to the list of assumptions made and not made in the SI.

(Main text Page 5 Line 93-95) “A complete list of assumptions underlying the construction of this model, including those that were deliberately excluded, is provided in the Supplementary Information (Extended Methods).”

2. Also, the

references <https://academic.oup.com/pcp/article/64/8/858/7148152>, <https://www.biorxiv.org/content/10.1101/2024.04.18.590023v2>

and <https://www.biorxiv.org/content/10.1101/2025.01.26.634902v1>) I mentioned talk about PSI as a quencher of PSII, not state transitions (as mentioned in the rebuttal letter). Although it is still not well documented if this is a mechanism of importance in angiosperms including *Nicotiana* (and there is no useful mutant), what would it mean for the conclusions if spillover is there?

Response: We appreciate the reviewer highlighting this emerging line of research again. The papers provided propose that the formation of PSI-PSII supercomplexes, along with the associated FRET-mediated excitation spillover from PSII to PSI, constitute a potential quenching mechanism, possibly involving both PsbS and zeaxanthin. In our model framework, such a spillover mechanism might potentially be captured by the PZ ↔ QZ interconversion or represented as part of qZ contribution, especially if PSI-PSII supercomplex formation correlates with zeaxanthin accumulation. We agree that further mechanistic insights and mutant tools will be valuable for evaluating the physiological significance of this process in different species. In response to this comment, we added the following sentence to the Introduction of the manuscript:

(Main text Page 4-5 Line 85-88) "Recent studies have also proposed the formation of PSI-PSII supercomplexes and associated excitation spillover from PSII to PSI as a potential quenching mechanism involving Zea and PsbS⁴⁰⁻⁴², though its physiological relevance in *N. benthamiana* remains to be fully established."

3. Anyway, I am still somewhat doubtful about the value of this contribution. Maybe I do not understand enough mathematics/biophysics to appreciate why this is suitable for Nature Communications, I feel that I do not really know more about NPQ after reading and thinking of the ms than I did before. If the other reviewers and the editor think differently I will not complain if it gets accepted, but I am not convinced that it is important enough...

Response: We appreciate the reviewer's honest feedback. Please also see the response to Comment 3 & 5 of Reviewer 1.

We agree that discovering new NPQ mechanisms is of great interest. However, one of our main motivations for studying NPQ is to ultimately improve crop yield. What is currently lacking in the field is a comprehensive framework that can isolate the contributions of each quenching component and construct a model with predictive power across different genotypes.

Therefore, the aim of this particular work is not to uncover new molecular players in NPQ, but rather to provide a quantitative modeling framework that enables kinetic interpretation of chlorophyll fluorescence lifetime-measured NPQ behavior under dynamic light conditions. This complements existing PAM-based approaches (the advantages of fluorescence lifetime measurements are discussed extensively in our previous response and in the Methods section of the manuscript) and helps bridge a long-standing methodological gap in NPQ kinetic analysis.

In the course of developing this model, we uncovered several notable findings and would like to highlight the most important ones again:

1. Quantification of the per molecule effectiveness of different xanthophylls in promoting quenching (Vio, Ant, Zea, and Lut)

These values, presented in Table 2, are not predictions but are directly obtained by fitting the model to experimental data. While debates exist regarding which carotenoid is the most effective (e.g., Lut vs Zea), our data provide a quantitative *in vivo* answer expressed in terms of effectiveness per molecule.

2. Time-resolved contribution of each NPQ component

As shown in Fig. 4 and Movie S1, we quantify the dynamical contribution of individual quenching components in various genotypes over illumination time. We emphasize the novelty of this dynamic perspective: the relative contributions of different NPQ components vary significantly across different time points and illumination phases. Again, these values are not merely predictions, but are extracted through robust fitting to high-quality experimental data. Given the reliability of our data and fits, we are considerably confident in the accuracy of these estimates and the conclusions drawn.

Ultimately, we believe the value of our work lies in offering a broadly applicable and extensible tool for dissecting NPQ contributions and dynamics *in vivo*. With further improvements in the future, we hope this model can become even more adaptable to natural illumination conditions and gain predictive capability regarding the expression levels of key proteins (as discussed in Fig. 5) to improve overall light-harvesting efficiency. In this way, our study provides a foundational step toward guiding future improvements in crop yield through more refined understanding and engineering of NPQ.

As is evident from this study, the work would not have been possible without a deep integration of expertise spanning plant biology, biochemistry, and genetics, as well as physical chemistry (spectroscopy and kinetic modeling). The highly interdisciplinary nature of this work bridging experimental and theoretical approaches is precisely why we hope to publish in *Nature Communications*. We believe that this platform offers an ideal venue for reaching a broad scientific audience, which can foster cross-disciplinary engagement. In that spirit, we have greatly appreciated the diverse perspectives reflected in the reviewers' comments, which align with our goal of encouraging dialogue across fields.

Reviewer #4 (Remarks to the Author):

1. The authors have engaged with my previous review in a constructive manner and addressed all the points I raised. As noted by the authors and raised by other reviewers, NPQ is an area in which there has been a lot of disagreement and argument over details. It is refreshing to see the approach here that attempts to take a broader view. The changes made by the authors improve, for me at least, the readability of the paper.

As noted in the previous review, I have only low expertise in the modelling aspects of the work, but assuming other referees are satisfied with this, I am happy to support publication.

Response: We thank the reviewer for recognizing our effort to adopt a broader and integrative approach to studying NPQ, especially in a field where mechanistic interpretations have often diverged. We are especially encouraged that the revisions have improved the readability of the manuscript from the reviewer's perspective.

Reviewer #4 (Remarks on code availability):

This aspect is outside my area of expertise.

Reviewer #5 (Remarks to the Author):

1. This is a revised manuscript that I evaluated already. Revisions took in due account criticisms and suggestions. Thus, the revised manuscript is presently suitable for publication and will constitute a valid tool for screening mutants for improved light use efficiency in different conditions. This is a valuable contribution for both basic and applied science.

Response: We thank the reviewer for their encouraging comments. It is gratifying to know that the revisions have addressed the earlier concerns and improved the manuscript's clarity. We appreciate the recognition that this work may serve as a useful tool for screening mutants to improve crop yield. We hope that our future development of this model will further enhance its applicability and ultimately contribute to achieving that goal.

2. Having said so, Authors might have considered the results they have obtained with the (new) genotype ZEP npq4 of minor importance with respect to the general importance of the ms which deals with modelling. Nevertheless, the reduced lifetime in the dark in the ZEP npq4 points to an autonomous activity of Zea in quenching that could well be highlighted in the discussion.

Response: We thank the reviewer for their careful review of the data and agree with their assessment regarding *npq4zep2*. In response, we have incorporated the following content into the Discussion.

(Main text Page 17 Line 387-391) “The significant qZ contributions in our model are consistent with the existence of a light-independent Zea-mediated quenching mechanism¹⁶. This interpretation is further supported by the observed reduction in the initial τ_{dark} in *npq4zep2* compared to *npq4*, suggesting a Δ pH-insensitive Zea-dependent quenching component that remains active in the dark-acclimated state in *npq4zep2*.”

Reviewer #5 (Remarks on code availability):

I did not review the code. But a co-worker in my lab has downloaded it and found it usable. It will take some time for a more thoughtful evaluation.

RESPONSE TO REVIEWER COMMENTS

For clarity, we have color-coded this response letter as follows:

Black: Reviewer comments

Blue: Our responses

Red: Direct citations of the manuscript (Line numbers refer to the pdf version of the final edits of the manuscript.)

Reviewer #1 (Remarks to the Author):

I appreciate the authors' detailed and thoughtful responses and acknowledge that the manuscript has been improved in several important respects, particularly with regard to clarification of modelling assumptions, expansion of the literature review, and more careful positioning of zeaxanthin's role in NPQ. However, despite these improvements, my central concerns remain unresolved, and I do not feel able to recommend publication of the manuscript in Nature Communications. Most importantly, the study still lacks experimental characterisation of electron transport activity and ΔpH generation in the specific thylakoid preparations used for the transient absorption and lifetime measurements. The authors rely on general literature and modelling assumptions to justify treating ΔpH as an instantaneous, constant-amplitude switch, but no direct or proxy measurements (proton gradient indicators or electron transport rates) are provided. Without such information, it is not possible to assess whether the kinetic parameters extracted from the model are biologically meaningful or comparable across genotypes. This limitation undermines confidence in both the experimental interpretation and the quantitative modelling. While the authors correctly state that their model is intended to be mechanistically agnostic and primarily kinetic, this does not remove the need for adequate physiological constraints on the system being modelled. In my view, the absence of ΔpH and electron transport characterisation remains a fundamental weakness, rather than a secondary or stylistic issue. The authors have also clarified their position regarding zeaxanthin and expanded their discussion of alternative models of NPQ, which is welcome. Nevertheless, the work still engages with long-standing and unresolved controversies in photosynthesis research in a way that, in my opinion, limits its accessibility and general appeal to the broad readership of Nature Communications. The study will likely be of significant interest to specialists in NPQ modelling and ultrafast spectroscopy, but its current scope and experimental grounding do not, in my view, justify publication in a general high-impact journal.

In summary, although I recognise the technical sophistication of the modelling approach and the effort invested in revision, I remain unconvinced that the manuscript has reached the level of experimental completeness, robustness, and general relevance required for Nature Communications. I therefore recommend decline, while noting that the work may be more suitable for a specialist journal in photosynthesis or biophysical modelling following further experimental validation.

Response: We are glad to hear that our manuscript has improved after the previous revision. Regarding the ΔpH concern, we now realize that there may be a misunderstanding that this work was performed using isolated thylakoids. Instead, **all** experimental data used in this manuscript, including chlorophyll fluorescence lifetime measurements (TCSPC) and pigment profiling by HPLC, were conducted in intact, freshly harvested whole leaves of *N. benthamiana* after sufficient dark acclimation. This is stated in the Abstract and Results and described in more detail in the Methods. To make this clearer, we have revised a few sentences in the Introduction, as follows.

(Main text Page 7 Line 139-140) “This choice could enable both kinetic and mechanistic analysis of NPQ in land plants when combined with complementary techniques in a series of studies. In the present study, however, to preserve physiological function as much as possible, we utilized fresh, intact leaves of *N. benthamiana* to systematically investigate the NPQ response using a comprehensive set of single and double NPQ mutants²⁶ (Table 1) across three regular/irregular actinic light sequences.”

(Main text Page 7 Line 146-147) “Model parameters are directly fitted to the experimental Chl fluorescence lifetimes, guided by xanthophyll concentrations measured via high-performance liquid chromatography (HPLC), with all measurements performed in intact leaves as described in more detail in the Methods.”

We apologize for any confusion caused by our mention of previous work^{26,57} using transient absorption spectroscopy as part of the reason we chose *N. benthamiana* as a model organism for a series of our NPQ studies, including this one. In the two previous studies we referenced^{26,57}, transient absorption spectroscopy required sending a laser pulse through the sample and detecting the transmitted signal, which makes whole-leaf measurements impractical due to strong scattering. We therefore were constrained to use isolated thylakoids in those transient absorption studies^{26,57}. However, we totally agree that some physiological functions, including ΔpH , can be affected by the thylakoid extraction process, and that measuring a more intact system (such as whole leaves) is preferred whenever possible. Accordingly, for the present study, because all measurements were conducted in whole leaves, we believe it is reasonable to treat ΔpH as a rapid switch compared to the timescale of the NPQ behaviors we quantify (minutes). We hope that, in future work, we can model more complicated ΔpH behavior and incorporate that into our model.

We respectfully maintain that the manuscript is appropriate for *Nature Communications*, as it presents a generalizable approach for disentangling photoprotective kinetics in intact leaves and demonstrates predictive power across genotypes and illumination regimes, with implications for optimizing photoprotection under fluctuating light and improving light-use efficiency—an area of broad interest beyond NPQ or modelling specialists.

Reviewer #3 (Remarks to the Author):

The ping-pong between authors and reviewers has improved the ms further, I have no specific comments on the presentation. I still think that this fits better in a specialized journal (and Reviewer 1 seems to agree) but if the decision will be to accept, OK with me.

Response: We thank the reviewer's constructive feedback throughout the revision process, which has helped us improve the manuscript further. With respect to whether the work is appropriate for *Nature Communications*; please refer to our comments to Reviewer 1. We are grateful that, should the editors judge the manuscript appropriate for *Nature Communications*, the reviewer would be supportive.

Reviewer #6 (Remarks to the Author):

In my opinion the assumptions of the model are appropriately justified. I think part of the problem lies in a misunderstanding between the reviewers and the authors. The reviewer writes: "Most importantly, the study still lacks experimental characterisation of electron transport activity and ΔpH generation in the specific thylakoid preparations used for the transient absorption and lifetime measurements." It seems the reviewer assumes the measurements are done on isolated thylakoids and that transient absorption has been used. This confusion arises from the introduction where this is written: "To that end, we selected *Nicotiana benthamiana* as our model system due to its compatibility with transient absorption spectroscopic measurements, which facilitate investigation of potential qE quenching mechanisms and assessment of exciton diffusion length in relation to NPQ levels in isolated thylakoids". However, in the manuscript only TCSPC measurements on whole leaves are reported, which is not mentioned in the introduction.

The electron transport activity and ΔpH generation would be a matter of concern in isolated systems, but are well documented in intact leaves. The reviewer is right that it will take some time to build the pH gradient, but the time-resolution in this study is not very high, so it should not be a limiting factor.

Response: We appreciate the reviewer's thoughtful assessment of our study and for noting that the primary concern raised by Reviewer 1 appears to stem from a misunderstanding that our measurements were performed on isolated thylakoids. As the reviewer pointed out, this confusion is likely due to the wording in our Introduction. To address this clearly, we have revised the Introduction; please refer to our comment to Reviewer 1. We thank the reviewer for noting that our model assumptions are appropriately justified, and that treating ΔpH as a rapid switch, despite not being modelled in full physiological detail, is a reasonable assumption given the timescale of NPQ behavior we studied.

I have one point which I would like to have clarified.

The authors write: "The memory effect explored by Matuszynska et al. is an intrinsic part of our model⁴³, as it was in our earlier algal model^{51,52337}, and is unraveled in greater mechanistic detail³³⁸ here, demonstrating the importance of the Ant intermediate to NPQ memory during dark periods." This is about the observation that NPQ is induced faster upon a second

illumination, which could be due to the presence of Ant or remaining Zea. However, in Figure S4 the faster NPQ induction is also observed for *npq1*, which cannot form Ant or Zea. The model still fits the data well, so there must be another explanation than remaining Ant/Zea.

Response: As the time series in Figure 4 and Supplementary Movie 1 show, Lut (q_{E_L}) is always the first responder when present. In genotypes with a functional VAZ cycle, as time of illumination progresses, the contributions from Zea (q_{E_Z} and q_Z) grow and become dominant by times longer than 5 minutes. In the *npq1* mutant, which lacks a VAZ cycle, the rapid initial response of Lut (q_{E_L}) becomes dominant (with a small contribution from Vio) during both the initial and subsequent illumination periods. This is presumably because Lut is constitutively present in LHC complexes and NPQ can be rapidly activated upon illumination. We also note that the relative contribution of Vio (q_{E_V}) is somewhat larger in *npq1* than in WT (e.g., Figure 4D).

These results highlight the importance of a time-resolved, dynamic picture when interpreting the contributions of different quenching species. To make our conclusions clearer, we updated the presentation of Supplementary Movie 1 (20HL) by adding background shading to indicate HL/dark periods, and added Supplementary Movie 2, which visualizes the dynamical contribution of each component under the 5HL-10D-5HL sequence. In particular, Movie S2 shows the rapid relaxation of q_{E_L} within ~ 1 min in darkness and its rapid re-initiation upon the second HL period. We also revised the Results and Discussion to emphasize the rapid activation of NPQ in *npq1* upon both the initial and re-illumination, and to provide additional clarification.

(Main text Page 11 Line 250-251) “In *npq1* (Fig. 2C), the NPQ components q_I , q_{E_V} , and q_{E_L} exhibited rapid activation, contributing to a sharp decrease in lifetime during both the initial and second illumination periods, although the model slightly underestimated Lut activation rates.”

(Main text Page 18-19 Line 423-424) “In *npq1*, lack of the VDE-mediated VAZ cycle results in increased Vio contribution to quenching, and Lut engages rapidly in quenching at the onset of illumination, consistent with our observations in WT.”

(Main text Page 19 Line 426-436) “Additionally, Supplementary Movie 2, which shows the same NPQ component decomposition under the 5HL-10D-5HL actinic light sequence, visualizes how these components contribute to the NPQ relaxation in darkness and NPQ re-initiation upon the second HL period. Interestingly, NPQ contributed by Lut (q_{E_L}) relaxes the fastest after the transition from HL to darkness, diminishing within ~ 1 min in WT, followed by q_{E_Z} . By contrast, q_Z decays much more slowly in darkness, aside from the slowest component, q_I . Notably, *npq1* shows faster NPQ re-initiation upon the second HL period (see Results and Supplementary Figure 4) despite the absence of Ant or Zea, supporting the rapid switching of q_{E_L} discussed above. These results highlight the importance of a time-resolved dynamic picture when interpreting the contributions of different quenching components, since they are highly time-dependent and strongly shaped by illumination duration and history.”